# Non-stationary Domain Generalization: Theory and Algorithm

**Thai-Hoang Pham[1,2]**          **Xueru Zhang[1]**          **Ping Zhang[1,2]**

[1]Department of Computer Science and Engineering, The Ohio State University, USA
[2]Department of Biomedical Informatics, The Ohio State University, USA
{pham.375,zhang.12807,zhang.10631}@osu.edu

## Abstract

Although recent advances in machine learning have shown its success to learn from independent and identically distributed (IID) data, it is vulnerable to out-of-distribution (OOD) data in an open world. Domain generalization (DG) deals with such an issue and it aims to learn a model from multiple source domains that can be generalized to unseen target domains. Existing studies on DG have largely focused on stationary settings with homogeneous source domains. However, in many applications, domains may evolve along a specific direction (e.g., time, space). Without accounting for such non-stationary patterns, models trained with existing methods may fail to generalize on OOD data. In this paper, we study domain generalization in non-stationary environment. We first examine the impact of environmental non-stationarity on model performance and establish the theoretical upper bounds for the model error at target domains. Then, we propose a novel algorithm based on adaptive invariant representation learning, which leverages the non-stationary pattern to train a model that attains good performance on target domains. Experiments on both synthetic and real data validate the proposed algorithm.

## 1 INTRODUCTION

Many machine learning (ML) systems are built based on an assumption that training and testing data are sampled independently and identically from the same distribution. However, this is commonly violated in real applications where the environment changes during model deployment, and there exist distribution shifts between training and testing data. The problem of training models that are robust under distribution shifts is typically referred to as domain adaptation (or generalization), where the goal is to train a model on *source* domain that can generalize well on a *target* domain. Specifically, domain adaptation (DA) aims to deploy model on a *specific* target domain, and it assumes the data from this target domain is accessible during training. In contrast, domain generalization (DG) considers a more realistic scenario where target domain data is unavailable during training; instead it leverages multiple source domains to learn models that generalize to *unseen* target domains.

For both DA and DG, various approaches have been proposed to learn a robust model with high performance on target domains. However, most of them assume both source and target domains are sampled from a *stationary* environment; they are not suitable for settings where the data distribution evolves along a specific direction (e.g., time, space). In stationary DG, the domains are treated as an unordered set, while in non-stationary DG, they form an ordered tuple with a sequential structure (see Figure 1). This defining characteristic of non-stationary DG renders this setting a challenging task, necessitating novel solutions that account for non-stationary mechanisms. In practice, evolvable data distributions have been observed in many applications. For example, satellite images change over time due to city development and climate change [Christie et al., 2018], clinical data evolves due to changes in disease prevalence [Guo et al., 2022], facial images gradually evolve because of the changes in fashion and social norms [Ginosar et al., 2015]. Without accounting for the non-stationary patterns across domains, existing methods in DA/DG designed for stationary settings may not perform well in non-stationary environments. As evidenced by Guo et al. [2022], clinical predictive models trained under existing DA/DG methods cannot perform better on future clinical data compared to empirical risk minimization.

In this paper, we study *domain generalization* (DG) in non-stationary environments. The goal is to learn a model from a sequence of source domains that can capture the non-stationary patterns and generalize well to (multiple) *unseen* target domains. We first examine the impacts of

non-stationary distribution shifts and study how the model performance attained on source domains can be affected when the model is deployed on target domains. Based on the theoretical findings, we propose an algorithm named Adaptive Invariant Representation Learning (`AIRL`); it minimizes the error on target domains by learning a sequence of representations that are *invariant* for every two consecutive source domains but are *adaptive* across these pairs.

In particular, `AIRL` consists of two components: (i) *representation network*, which is trained on the sequence of source domains to learn invariant representations between every two consecutive source domains, (ii) *classification network* that minimizes the prediction errors on source domains. Our main idea is to create adaptive representation and classification networks that can evolve in response to the dynamic environment. In other words, we aim to find networks that can effectively capture the non-stationary patterns from the sequence of source domains. At the inference stage, the representation network is used to generate the optimal representation mappings and the classification network is used to make predictions in the target domains, without the need to access their data. To verify the effectiveness of `AIRL`, we conduct extensive experiments on both synthetic and real data and compare `AIRL` with various existing methods.

## 2 RELATED WORK

This work is closely related to the literature on domain generalization, continuous (or gradual) domain adaptation, continual learning. We introduce each topic and discuss their differences with our work.

**Domain generalization.** The goal is to learn a model on multiple source domains that can generalize to the out-of-distribution samples from an unseen target domain. Depending on the learning strategy, existing works for DG can be roughly classified into three categories: (i) methods based on *domain-invariant representation learning* [Phung et al., 2021, Nguyen et al., 2021a, Pham et al., 2023]; (ii) methods based on *data manipulation* [Qiao et al., 2020, Zhou et al., 2020]; (iii) methods by considering DG in general ML paradigms and using approaches such as *meta-learning* [Li et al., 2018a, Balaji et al., 2018], *gradient operation* [Rame et al., 2021, Tian et al., 2022], *self-supervised learning* [Jeon et al., 2021, Li et al., 2021], and *distributional robustness* [Koh et al., 2021, Wang et al., 2021]. However, these works assume both source and target domains are sampled from a *stationary* environment and they do not consider the non-stationary patterns across domains; this differs from our setting.

**Non-stationary domain generalization.** To the best of our knowledge, only a few concurrent works study domain generalization in non-stationary environments [Bai et al., 2022, Qin et al., 2022, Zeng et al., 2023, Xie et al., 2024, Zeng

et al., 2024b,a]. However, the problem settings considered in these works are rather limited. For example, Qin et al. [2022] only focuses on the environments that evolve based on a *consistent* and *stationary* transition function; the approaches in Bai et al. [2022], Zeng et al. [2023, 2024b] can only generalize the model to a *single subsequent* target domain; Qin et al. [2022], Xie et al. [2024], Zeng et al. [2024a] assume that data are aligned across domain sequence. In contrast, this paper considers a more general setting where data may evolve based on non-stationary dynamics, and the proposed algorithm learned from the sequence of unaligned source domains can generate models for multiple unseen target domains.

**Continuous domain adaptation.** Unlike conventional DA/DG methods that only consider categorical domain labels, continuous DA admits continuous domain labels such as space, time [Ortiz-Jimenez et al., 2019, Wang et al., 2020]. Specifically, this line of research considers scenarios where the data distribution changes gradually and domain labels are continuous. Similar to conventional DA, samples from target domain are required to guide the model adaptation process. This is in contrast to this study, which considers the target domains whose samples are inaccessible during training.

**Gradual domain adaptation.** Similar to continuous DA, Gradual DA also considers continuous domain labels, and the samples from the target domain are accessible during training [Kumar et al., 2020, Chen et al., 2020, Chen and Chao, 2021]. The prime difference is that continuous DA focuses on the generalization from a single source domain to a target domain, whereas there are multiple source domains in gradual DA.

**Continual learning.** The goal is to learn a model continuously from a sequence of tasks. The main focus in continual learning is to overcome the issue of catastrophic forgetting, i.e., prevent forgetting the old knowledge as the model is learned on new tasks [Chaudhry et al., 2018, Kirkpatrick et al., 2017, Mallya and Lazebnik, 2018]. This differs from temporal-shift DG (i.e., a special case of our setting) which aims to train a model that can generalize to future domains.

## 3 PROBLEM FORMULATION

We first introduce the notations used throughout the paper and then formulate the problem. These notations and their descriptions are also summarized in Table 1.

**Notations.** Let $\mathcal{X}$ and $\mathcal{Y}$ denote the input and output space, respectively. We use capitalized letters $X, Y$ to denote random variables that take values in $\mathcal{X}, \mathcal{Y}$ and small letters $x, y$ their realizations. A *domain* $D$ is specified by distribution $P_D^{X,Y} : \mathcal{X} \times \mathcal{Y} \to [0,1]$ and labeling function $\mathbb{h}_D : \mathcal{X} \to \mathcal{Y}^\Delta$, where $\Delta$ is a probability simplex over $\mathcal{Y}$. For simplicity, we also use $P_D^V$ (or $P_D^{V|U}$) to denote the

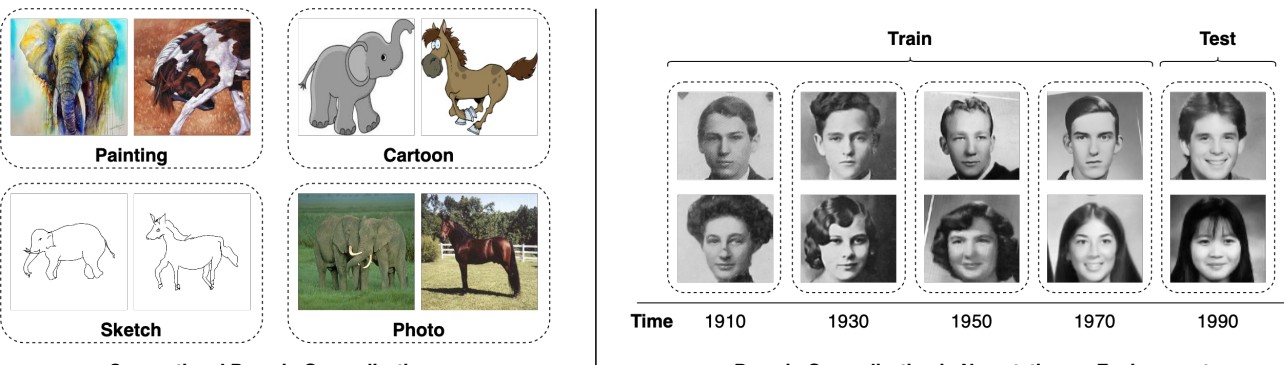

Figure 1: An illustrative comparison between conventional DG and DG in non-stationary environment: domains in conventional DG are independently sampled from a stationary environment, whereas DG in non-stationary environment considers domains that evolve along a specific direction. As shown in the right plot, data (i.e., images) changes over time and the model trained on past data may not have good performance on future data due to non-stationarity (i.e., temporal shift).

induced marginal (or conditional) distributions of random variable $V$ (given $U$) in the domain $D$.

**Non-stationary domain generalization setup.** We consider a problem where a learning algorithm has access to sequence of source datasets $\{S_t\}_{t=1}^{T}$ where $S_t$ consists of $n$ instances i.i.d sampled from source domain $D_t$. In non-stationary DG, we assume there exists a mechanism $\mathbb{M}$ that captures non-stationary patterns in the data. Specifically, $\mathbb{M}$ can generate a sequence of mapping functions $\{\mathrm{m}_t\}_{t\in\mathbb{N}}$ in which $\mathrm{m}_t : \mathcal{X} \times \mathcal{Y} \to \mathcal{X} \times \mathcal{Y}$ captures the transition from domain $D_{t-1}$ to domain $D_t$. In other words, we can regard $P_{D_t}^{X,Y}$ as the push-forward distribution induced from $P_{D_{t-1}}^{X,Y}$ using the mapping function $\mathrm{m}_{t-1}$ (i.e., $P_{D_t}^{X,Y} := \mathrm{m}_{t-1}\sharp P_{D_{t-1}}^{X,Y}$). Note that this setup is different from conventional DG where domains are sampled independently from a meta-distribution. In non-stationary DG, domains are related to each other via mechanism $\mathbb{M}$ (i.e., $\mathrm{m}_t$ depends on previous mappings $\mathrm{m}_1, \mathrm{m}_2, \cdots, \mathrm{m}_{t-1}$).

Given a sequence of $T$ source domains, our goal is to learn a sequence of models $H = \{h_t\}_{t=T+1}^{T+K}$, where $h_t : \mathcal{X} \to \mathcal{Y}^\Delta$ in a hypothesis class $\mathcal{H}$ is a model corresponds to domain $D_t$, such that these models can perform well on $K$ (unseen) target domains $\{D_t\}_{t=T+1}^{T+K}$. We aim to investigate under what conditions and by what algorithms we can ensure models learned from source domains can attain high accuracy at unknown target domains $\{D_t\}_{t=T+1}^{T+K}$ in non-stationary environment. Formally, we measure the accuracy using an error metric defined below.

**Error metric.** Consider a model $h : \mathcal{X} \to \mathcal{Y}^\Delta$ in a hypothesis class $\mathcal{H}$, we denote $h(x)_y$ as the element on $y$-th dimension which predicts $\Pr(Y = y|X = x)$. Then the *expected error* of $h$ under domain $D$ for some loss function $L : \mathcal{Y}^\Delta \times \mathcal{Y} \to \mathbb{R}_+$ (e.g., 0-1, cross-entropy loss) can be defined as $\epsilon_D(h) = \mathbb{E}_{x,y\sim D}[L(h(X), Y)]$. Similarly, the *empirical error* of $h$ over $n$ samples $S$ drawn i.i.d.

from $P_D^{X,Y}$ is defined as $\epsilon_S(h) = \frac{1}{n}\sum_{x,y\in S} L(h(x), y)$. We also denote a family of functions $\mathcal{L}_\mathcal{H}$ associated with loss function $L$ and hypothesis class $\mathcal{H}$ as $\mathcal{L}_\mathcal{H} = \{(x,y) \to L(h(x), y) : h \in \mathcal{H}\}$.

Non-stationary mechanism $\mathbb{M}$ is a key component in non-stationary DG. Since $\mathbb{M}$ is unknown, we need to learn it from source domains. Let $M$ be a learned mechanism in a hypothesis class $\mathcal{M}$. To support theoretical analysis about $M$, we define the following:

- $M$-**generated domain sequence** $\{D_1^M, \cdots, D_{T+K}^M\}$ where domain $D_t^M$ is associated with the distribution $P_{D_t^M}^{X,Y} = m_{t-1}\sharp P_{D_{t-1}^M}^{X,Y}$ and $D_1^M = D_1$.
- $M$-**generated dataset sequence** $\{S_1^M, \cdots, S_{T+K}^M\}$ where dataset $S_t^M$ is generated from dataset $S_{t-1}$ by using mapping $m_{t-1}$.
- $M$-**optimal model sequence** $H^M = \{h_1^M, \cdots, h_{T+K}^M\}$ where $h_t^M = \arg\min_{h\in\mathcal{H}} \epsilon_{D_t^M}(h)$.
- $M$-**empirical optimal model sequence** $\widehat{H}^M = \{\widehat{h}_1^M, \cdots, \widehat{h}_{T+K}^M\}$ where $\widehat{h}_t^M = \arg\min_{h\in\mathcal{H}} \epsilon_{S_t^M}(h)$.

We also denote errors of model sequence $H$ on source and target domains as $E_{src}(H) = \frac{1}{T}\sum_{t=1}^{T} \epsilon_{D_t}(h_t)$ and $E_{tgt}(H) = \frac{1}{K}\sum_{t=T+1}^{T+K} \epsilon_{D_t}(h_t)$, respectively, and on $M$-generated source and target domains as $E_{src}^M(H) = \frac{1}{T}\sum_{t=1}^{T} \epsilon_{D_t^M}(h_t)$ and $E_{tgt}^M(H) = \frac{1}{K}\sum_{t=T+1}^{T+K} \epsilon_{D_t^M}(h_t)$. Empirical errors of $H$ on source and target datasets ($\widehat{E}_{src}$ and $\widehat{E}_{tgt}$), and on $M$-generated source and target datasets ($\widehat{E}_{src}^M$ and $\widehat{E}_{tgt}^M$) are defined similarly (Table 1).

## 4 THEORETICAL RESULTS

In this section, we aim to understand how a model sequence $H$ learned from source domain data would perform when deployed in target domains under non-stationary distribution

Table 1: Notations used in this paper.

| Notation | Description |
|---|---|
| $\mathcal{X}, \mathcal{Y}, \mathcal{Z}$ | input, output, representation spaces |
| $\mathcal{M}, \mathcal{H}$ | mechanism and model hypothesis classes |
| $X, Y, Z$ (resp. $x, y, z$) | random variables (resp. realizations) in $\mathcal{X}, \mathcal{Y}, \mathcal{Z}$ |
| $D_t$ | $t^{th}$ domain in domain sequence |
| $S_t$ | $t^{th}$ dataset sampled from domain $D_t$ |
| $\{D_t\}_{t=1}^{T}$ | source domains |
| $\{D_t\}_{t=T+1}^{T+K}$ | target domains |
| $P_{D_t}^{X,Y}$ | distribution associated with domain $D_t$ |
| $\mathbb{h}_{D_t}: \mathcal{X} \to \mathcal{Y}^\Delta$ | labeling function of domain $D_t$ |
| $\mathbb{M}$ | ground-truth mechanism that generates $\{\mathbb{m}_t\}_{t \in \mathbb{N}}$ |
| $\mathbb{m}_t$ | ground-truth mapping from $D_t$ to $D_{t+1}$: $P_{D_{t+1}}^{X,Y} = \mathbb{m}_t \sharp P_{D_t}^{X,Y}$ |
| $M \in \mathcal{M}$ | hypothesis mechanism that generates $\{m_t\}_{t \in \mathbb{N}}$ |
| $h_t \in \mathcal{H}$ | hypothesis classifier for domain $D_t$ |
| $L: \mathcal{Y}^\Delta \to \mathcal{Y}$ | loss function |
| $\mathcal{L}_\mathcal{H}$ | family of functions $\{(x,y) \to L(h(x),y) : h \in \mathcal{H}\}$ |
| $\epsilon_D(h)$ | expected error of classifier $h$ on domain $D$ |
| $\epsilon_S(h)$ | empirical error of classifier $h$ on dataset $S$ |
| $D_t^M$ | domain associated with distribution $P_{D_t^M}^{X,Y} = m_{t-1} \sharp P_{D_{t-1}}^{X,Y}$ |
| $S_t^M$ | dataset associated with domain $D_t^M$ |
| $h_t^M$ | $\arg\min_{h \in \mathcal{H}} \epsilon_{D_t^M}(h)$ |
| $\widehat{h}_t^M$ | $\arg\min_{h \in \mathcal{H}} \epsilon_{S_t^M}(h)$ |
| $\mathcal{D}_{JS}$ | JS-divergence |
| $E_{src}(H)$ (resp. $E_{tgt}(H)$) | $\frac{1}{T}\sum_{t=1}^{T} \epsilon_{D_t}(h_t)$ (resp. $\frac{1}{K}\sum_{t=T+1}^{T+K} \epsilon_{D_t}(h_t)$) |
| $E_{src}^M(H)$ (resp. $E_{tgt}^M(H)$) | $\frac{1}{T}\sum_{t=1}^{T} \epsilon_{D_t^M}(h_t)$ (resp. $\frac{1}{K}\sum_{t=T+1}^{T+K} \epsilon_{D_t^M}(h_t)$) |
| $\widehat{E}_{src}(H)$ (resp. $\widehat{E}_{tgt}(H)$) | $\frac{1}{T}\sum_{t=1}^{T} \epsilon_{S_t}(h_t)$ (resp. $\frac{1}{K}\sum_{t=T+1}^{T+K} \epsilon_{S_t}(h_t)$) |
| $\widehat{E}_{src}^M(H)$ (resp. $\widehat{E}_{tgt}^M(H)$) | $\frac{1}{T}\sum_{t=1}^{T} \epsilon_{S_t^M}(h_t)$ (resp. $\frac{1}{K}\sum_{t=T+1}^{T+K} \epsilon_{S_t^M}(h_t)$) |
| $D_{src}(M)$ | $\frac{1}{T}\sum_{t=1}^{T}\left(\mathcal{D}_{JS}\left(P_{D_t}^{X,Y} \parallel P_{D_t^M}^{X,Y}\right)\right)^{1/2}$ |
| $D_{tgt}(M)$ | $\frac{1}{K}\sum_{t=T+1}^{T+K}\left(\mathcal{D}_{JS}\left(P_{D_t}^{X,Y} \parallel P_{D_t^M}^{X,Y}\right)\right)^{1/2}$ |
| $\Phi(\mathcal{M}, \mathcal{H})$ | $\sup_{M' \in \mathcal{M}}\left(E_{tgt}\left(H^{M'}\right) - E_{src}\left(H^{M'}\right)\right)$ |
| $\Phi(\mathcal{M})$ | $\sup_{M' \in \mathcal{M}}\left(D_{tgt}(M') - D_{src}(M')\right)$ |

shifts. Specifically, we will develop theoretical upper bounds of the model sequence's errors at target domains. These theoretical findings will provide guidance for the algorithm design in Section 5. All proofs are in Appendix A.

To start, we adopt two assumptions commonly used in DA/DG literature [Nguyen et al., 2021b, Kumar et al., 2020].

**Assumption 1** (Bounded loss). We assume loss function $L$ is upper bounded by a constant $C$, i.e., $\forall x \in \mathcal{X}, y \in \mathcal{Y}$, $h \in \mathcal{H}$, we have $L(h(x), y) \leq C$.

**Assumption 2** (Bounded model complexity). We assume Rademacher complexity [Bartlett and Mendelson, 2002] of function class $\mathcal{L}_\mathcal{H}$ computed from all samples with size $n$ is bounded for any distribution $P$ considered in this paper. That is, for some constant $B > 0$, we have:

$$\mathcal{R}_n(\mathcal{L}_\mathcal{H}) = \mathbb{E}\left[\sup_{f \in \mathcal{L}_\mathcal{H}} \frac{1}{n}\sum_{i=1}^{n} \sigma_i f(x_i)\right] \leq \frac{B}{\sqrt{n}}$$

where the expectation is with respect to $x_i \sim P$ and $\sigma_i \sim P_\mathcal{R}$, and $P_\mathcal{R}$ is Rademacher distribution.

We note that these two assumptions are actually reasonable and not strong. For instance, although Assumption 1 does not hold for cross-entropy loss used in classification, we can modify this loss to make it satisfied Assumption 1. In particular, it can be bounded by $C$ by modifying softmax

output from $(p_1, \cdots, p_{|\mathcal{Y}|})$ to $(\hat{p}_1, \cdots, \hat{p}_{|\mathcal{Y}|})$ where $\hat{p}_i = p_i(1 - \exp(-C)|\mathcal{Y}|) + \exp(-C)$. In addition, according to Liang [2016] (Theorem 11 page 82), Assumption 2 holds when input space is compact and bounded in unit $L_2$ ball and function $f$ in $\mathcal{L}_\mathcal{H}$ is linear and Lipschitz continuous in $l_2$ norm.

To learn a model sequence $H$ that performs well on unseen target domains, we need to account for the non-stationary patterns across domains. However, these patterns are governed by mechanism $\mathbb{M}$ which is unknown and must be estimated from source domains. Therefore, we need to learn a mechanism $M \in \mathcal{M}$ that can well estimate ground-truth $\mathbb{M}$ and learn $H$ by leveraging $M$. Because the target data is inaccessible, we expect that the model performance on the target highly depends on the accuracy of $M \in \mathcal{M}$. To formally characterize the complexity of learning non-stationary pattern leveraging hypothesis classes $\mathcal{M}, \mathcal{H}$ and source domains, we introduce two complexity terms as follows.

**Definition 1** (**Non-stationary complexity**). Given a sequence of domains $\{D_t\}_{t=1}^{T+K}$, hypothesis classes $\mathcal{M}$ and $\mathcal{H}$, the $\mathcal{M}, \mathcal{H}$-complexity term $\Phi(\mathcal{M}, \mathcal{H})$ and $\mathcal{M}$-complexity term $\Phi(\mathcal{M})$ are defined as

$$\Phi(\mathcal{M}, \mathcal{H}) = \sup_{M' \in \mathcal{M}}\left(E_{tgt}\left(H^{M'}\right) - E_{src}\left(H^{M'}\right)\right)$$
$$\Phi(\mathcal{M}) = \sup_{M' \in \mathcal{M}}\left(D_{tgt}(M') - D_{src}(M')\right)$$

where $D_{tgt}(M') = \frac{1}{K}\sum_{t=T+1}^{T+K}\left(\mathcal{D}_{JS}\left(P_{D_t}^{X,Y} \parallel P_{D_t^{M'}}^{X,Y}\right)\right)^{1/2}$, $D_{src}(M') = \frac{1}{T}\sum_{t=1}^{T}\left(\mathcal{D}_{JS}\left(P_{D_t}^{X,Y} \parallel P_{D_t^{M'}}^{X,Y}\right)\right)^{1/2}$, and $\mathcal{D}_{JS}(\cdot \parallel \cdot)$ is JS-divergence between two distributions.

In essence, $\Phi(\mathcal{M}, \mathcal{H})$ quantifies the gap between the source and target domains in terms of prediction errors of model sequence $H^M$. Meanwhile, $\Phi(\mathcal{M})$ evaluates the disparity in performance of $M$ regarding its ability to estimate non-stationary patterns in source and in target domain sequences. Performance is measured by the statistical distance between ground-truth and the distributions induced by $M$. Inspired by discrepancy measures used to quantity the differences between distributions [Mansour et al., 2009, Mohri and Muñoz Medina, 2012] $\Phi(\mathcal{M}, \mathcal{H})$ and $\Phi(\mathcal{M})$ explicitly take into account the hypothesis classes $\mathcal{M}$ and $\mathcal{H}$, and loss function $L$. This ensures that the bound constructed from these terms is directly related to the learning problem at hand. Next, we present a guarantee on target domains for $M$-empirical optimal model sequence $\widehat{H}^M$ as follows.

**Theorem 1.** *Given domain sequence $\{D_t\}_{t=1}^{T+K}$, dataset sequence $\{S_t\}_{t=1}^{T+K}$ sampled from $\{D_t\}_{t=1}^{T+K}$, for any $M \in \mathcal{M}$ ($M$ can depend on $\{S_t\}_{t=1}^{T+K}$) and any $0 < \delta < 1$, with probability at least $1 - \delta$ over the choice of dataset sequence*

$\{S_t\}_{t=1}^{T+K}$, we have:

$$E_{tgt}\left(\widehat{H}^M\right) \leq \widehat{E}_{src}^M\left(\widehat{H}^M\right) + 5\sqrt{2}C \times D_{src}(M)$$
$$+ \Phi(\mathcal{M}) + 2\sqrt{2}C \times \Phi(\mathcal{M}, \mathcal{H})$$
$$+ \frac{6B}{\sqrt{n}} + 3\sqrt{\frac{\log((T+K)/\delta)}{2n}}$$

It states that the expected error of $\widehat{H}^M$ on target domains $E_{tgt}\left(\widehat{H}^M\right)$ is upper bounded by four parts: (i) empirical error of $\widehat{H}^M$ on $M$-generated source datasets $\widehat{E}_{src}^M\left(\widehat{H}^M\right)$, (ii) the average distance between source datasets and $M$-generating source datasets $D_{src}(M)$, (iii) non-stationary complexity terms $\Phi(\mathcal{M})$ and $\Phi(\mathcal{M}, \mathcal{H})$, (iv) sample complexity term $\frac{6B}{\sqrt{n}} + 3\sqrt{\frac{\log((T+K)/\delta)}{2n}}$. We note that both the third and fourth parts remain fixed given hypothesis classes $\mathcal{M}$ and $\mathcal{H}$, and the sample size $n$ for each dataset in the sequence. It is also noteworthy that this bound still holds when $M$ depends on dataset sequence $\{S_t\}_{t=1}^{T+K}$, thereby allowing us to apply this bound for $M$ learned from $\{S_t\}_{t=1}^{T+K}$. In addition, $\widehat{E}_{src}^M\left(\widehat{H}^M\right) = \min_H \widehat{E}_{src}^M(H)$ by definition. Therefore, to minimize the expected error of $\widehat{H}^M$ on target domains, Theorem 1 suggests us to find a mechanism $M^* = \arg\min_{M \in \mathcal{M}} D_{src}(M)$ from source datasets $\{S_t\}_{t=1}^{T+K}$, and then learn model sequence $\widehat{H}^{M^*}$ that minimizes empirical error on $M^*$-generated dataset sequence.

Learning $M^*$ requires the model to find the optimal mapping $m_{t-1}^* : \mathcal{X} \times \mathcal{Y} \to \mathcal{X} \times \mathcal{Y}$ that minimizes the distance of the joint distributions $\mathcal{D}_{JS}\left(P_{D_t}^{X,Y} \| P_{D_t^{M^*}}^{X,Y}\right)$ for all $t \in \{1, \cdots, T\}$. To this end, we first minimize the distance of output distribution between the two domains $D_t, D_{t-1}$, then find an optimal mapping function in input space $\mathcal{X}$. That is, minimizing the distance of joint distributions in output and input space separately. This approach is formally stated in Proposition 1 below.

**Proposition 1.** *Let $P_{D_{t-1}^W}^{X,Y}$ be the distribution induced from $P_{D_{t-1}}^{X,Y}$ by importance weighting with factors $\{w_y\}_{y \in \mathcal{Y}}$ where $w_y = P_{D_t}^{Y=y}/P_{D_{t-1}}^{Y=y}$ (i.e., $P_{D_{t-1}^W}^{X=x,Y=y} = w_y \times P_{D_{t-1}}^{X=x,Y=y}$). Then for any mechanism $M$ that generates $\{m_t : \mathcal{X} \to \mathcal{X}\}_{t \in \mathbb{N}}$, we have the following:*

$$\mathcal{D}_{JS}\left(P_{D_t}^{X,Y} \| P_{D_t^{W,M}}^{X,Y}\right) = \mathbb{E}_{y \sim P_{D_t}^Y}\left[\mathcal{D}_{JS}\left(P_{D_t}^{X|Y} \| P_{D_t^{W,M}}^{X|Y}\right)\right]$$

*where $P_{D_t^{W,M}}^{X,Y} = m_{t-1} \sharp P_{D_t^W}^{X,Y}$ is a push-forward distribution induced from $P_{D_t^W}^{X,Y}$ using $m_{t-1}$.*

Proposition 1 suggests 2-step approach to learn $m_t : \mathcal{X} \times \mathcal{Y} \to \mathcal{X} \times \mathcal{Y}$: (i) reweight $P_{D_{t-1}}^{X,Y}$ with factors $\{w_y\}_{y \in \mathcal{Y}}$ (i.e., to minimize the distance of output distribution between

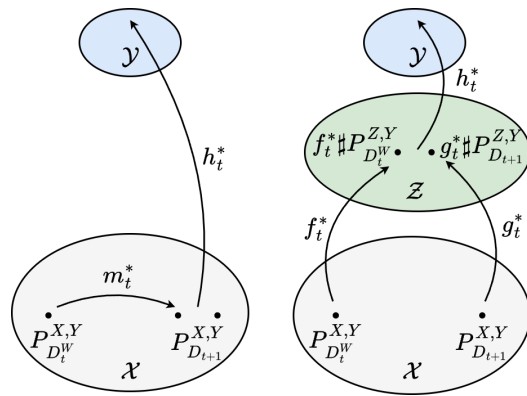

Figure 2: Visualization of learning non-stationary mapping between two domains $D_t^W$ (i.e., generated from $D_{t+1}$ by importance weighting) and $D_{t+1}$. (a) Learning in input space $\mathcal{X}$. (b) Learning in representation space $\mathcal{Z}$.

$D_t, D_{t-1}$); (ii) learn $m_t : \mathcal{X} \to \mathcal{X}$ that minimizes the distance of conditional distribution $\mathcal{D}_{JS}\left(P_{D_t}^{X|Y} \| P_{D_t^{W,M}}^{X|Y}\right)$.

We note that while the non-stationary complexity terms $\Phi(\mathcal{M})$ and $\Phi(\mathcal{M}, \mathcal{H})$ are fixed given hypothesis classes $\mathcal{M}$ and $\mathcal{H}$, a good design of $\mathcal{M}$ and $\mathcal{H}$ will make these terms small. Since the input space $\mathcal{X}$ may be of high dimension, constructing these hypothesis classes in high-dimensional space can be challenging in practice. To tackle this issue, we leverage the *representation learning* approach to first map inputs to a representation space $\mathcal{Z}$, which often has a lower dimension than $\mathcal{X}$. In particular, instead of using $m_t^* : \mathcal{X} \to \mathcal{X}$ to map $P_{D_t^W}^X$ to $P_{D_{t+1}^{W,M^*}}^X$ in input space $\mathcal{X}$, we use $f_t^* : \mathcal{X} \to \mathcal{Z}$ and $g_t^* : \mathcal{X} \to \mathcal{Z}$ to map $P_{D_t^W}^X$ and $P_{D_{t+1}}^X$ to $f_t^* \sharp P_{D_t^W}^Z$ and $g_t^* \sharp P_{D_{t+1}}^Z$ in representation space $\mathcal{Z}$ such that $\mathbb{E}\left[\mathcal{D}_{JS}\left(g_t^* \sharp P_{D_{t+1}}^{Z|Y} \| f_t^* \sharp P_{D_t^W}^{Z|Y}\right)\right]$ is minimal. Then, we learn a sequence of classifiers $H^*$ from representation to output spaces that minimizes empirical errors on source domains. This representation learning-based method is visualized in Figure 2 and is summarized below.

**Remark 1** (Representation learning). Given the sequence of $T$ source domains, we estimate:
(i) Non-stationary mechanism $F^*$ and $G^*$ that generate two sequence of representation mappings $\{f_t^* : \mathcal{X} \to \mathcal{Z}\}$ and $\{g_t^* : \mathcal{X} \to \mathcal{Z}\}$ with $F^*, G^*$ defined as:

$$\arg\min_{F \in \mathcal{F}, G \in \mathcal{G}} \frac{1}{T} \sum_{t=1}^{T} \mathbb{E}_{y \sim P_{D_t}^Y}\left[\mathcal{D}_{JS}\left(g_{t-1} \sharp P_{D_t}^{Z|Y} \| f_{t-1} \sharp P_{D_{t-1}^W}^{Z|Y}\right)\right]$$

where $F$ and $G$ generate sequence of representation mappings $\{f_t : \mathcal{X} \to \mathcal{Z}\}$ and $\{g_t : \mathcal{X} \to \mathcal{Z}\}$, $\mathcal{F}$ and $\mathcal{G}$ are the hypothesis classes of $F$ and $G$.
(ii) Sequence of classifiers $H^* = \{h_t^* : \mathcal{Z} \to \mathcal{Y}^\Delta\}$ where each $h_t^*$ minimizes the empirical errors with respect to dis-

tributions $f_t^* \sharp P_{D_t^W}^{Z,Y}$ and $g_t^* \sharp P_{D_{t+1}}^{Z,Y}$.

**Remark 2** (Comparison with conventional DG). A key property of non-stationary DG is that the model needs to evolve over the domain sequence to capture non-stationary patterns (i.e., learn invariant representations between two consecutive domains but adaptive across domain sequence). This differs from the conventional DG [Ganin et al., 2016, Phung et al., 2021] which (implicitly) assumes that target domains lie on or are near the mixture of source domains, then enforcing fixed invariant representations across all source domains can help generalize the model to target. We argue that this assumption does not hold in non-stationary DG where the target domains may be far from the mixture of source domains. Thus, the existing methods developed for conventional DG often fail in non-stationary DG. We further validate this empirically in Appendix C.2.

According to Remark 1, JS-divergence between two distribution $P_{D_t^W}$ and $P_{D_{t+1}}$ can be minimized through invariant representation learning. However in practice, models only have access to finite datasets $S_t^W$ and $S_{t+1}$. Moreover, Goodfellow et al. [2014] has shown that minimizing JS-divergence is aligned with the objective adversarial learning in the setting of infinite data. Therefore, evaluating the performance of minimizing JS-divergence via adversarial learning in the case of finite data is important. First, definition of adversarial learning is given below.

**Definition 2. Adversarial learning for invariant representation.** Given two datasets $S_t^w = \{x_t^i\}_{i=1}^n$ and $S_{t+1} = \{x_{t+1}^i\}_{i=1}^n$, the goal of adversarial learning approach for invariant representation with respect to these two datasets is to achieve $\widehat{L}_{adv}^t = \inf_{\alpha_t, \beta_t} \sup_{\gamma_t} \left( \frac{1}{n} \sum_{i=1}^n \log \left( D_{\gamma_t}(F_{\alpha_t}(x_t^i)) \right) + \frac{1}{n} \sum_{i=1}^n \log \left( 1 - D_{\gamma_t}(G_{\beta_t}(x_{t+1}^i)) \right) \right)$ where $F_{\alpha_t} : \mathcal{X} \to \mathcal{Z}$ and $G_{\beta_t} : \mathcal{X} \to \mathcal{Z}$ are the representation networks parameterized by $\alpha_t \in \mathcal{A}$ and $\beta_t \in \mathcal{B}$, and $D_{\gamma_t} : \mathcal{Z} \to [0,1]$ are the discriminator parameterized by $\gamma_t \in \Gamma$ that tries to predict which domain the representation comes from.

Then, Proposition 2 shows that the error of minimizing JS-divergences using adversarial learning on the sequence of source datasets size $n$ is up to $\mathcal{O}\left(\frac{1}{\sqrt{n}}\right)$.

**Proposition 2.** Let $\alpha_t^*, \beta_t^*, \gamma_t^*$ are parameters learned by infinite data and $\widehat{\alpha}_t, \widehat{\beta}_t, \widehat{\gamma}_t$ are parameters learned by optimizing $\widehat{L}_{adv}^t$, then we have:

$$\mathbb{E}\left[ D_{src}\left(\widehat{\alpha}, \widehat{\beta}\right) \right] \leq D_{src}\left(\alpha^*, \beta^*\right)$$
$$+ \mathcal{O}\left( \left(\frac{1}{\sqrt{n}}\right) \times C(\mathcal{A}, \mathcal{B}, \Gamma) \right)$$

where $D_{src}\left(\widehat{\alpha}, \widehat{\beta}\right) = \frac{1}{T} \sum_{t=1}^T \mathcal{D}_{JS}\left( P_{\widehat{\alpha}_t}^Z \parallel P_{\widehat{\beta}_t}^Z \right)$ and $D_{src}\left(\alpha^*, \beta^*\right) = \frac{1}{T} \sum_{t=1}^T \mathcal{D}_{JS}\left( P_{\alpha_t^*}^Z \parallel P_{\beta_t^*}^Z \right)$, $P_{\widehat{\alpha}_t}^Z$, $P_{\widehat{\beta}_t}^Z$,

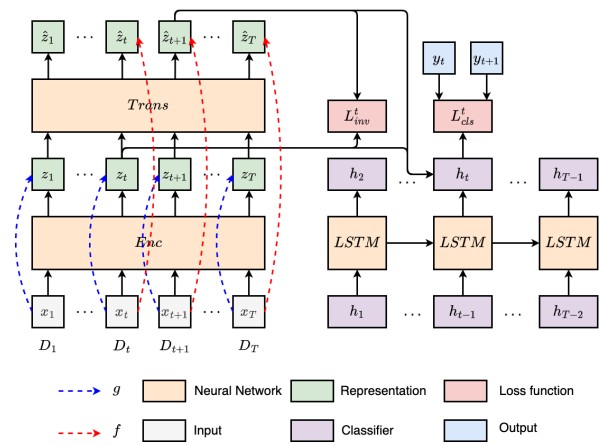

Figure 3: Overall architecture of `AIRL` and the visualization of its learning process.

$P_{\alpha_t^*}^Z$, $P_{\beta_t^*}^Z$ are distributions induced by representation networks parameterized by $\widehat{\alpha}_t, \widehat{\beta}_t, \alpha_t^*, \beta_t^*$, respectively, and $C(\mathcal{A}, \mathcal{B}, \Gamma)$ is a constant specified by the parameter spaces $\mathcal{A}, \mathcal{B}, \Gamma$.

# 5 PROPOSED ALGORITHM

**Overview.** Based on Remark 1, we propose `AIRL`, a novel model that learns adaptive invariant representations from a sequence of $T$ source domains. `AIRL` includes two components: (i) *representation network* which are instantiation of mechanisms $F^*$ and $G^*$ that generates representation mapping sequences $[f_1^*, \cdots, f_{T+K}^*]$ and $[g_1^*, \cdots, g_{T+K}^*]$ from input space to representation space, (ii) *classification network* that learns the sequence of classifiers $H^* = [h_1^*, \cdots, h_{T+K}^*]$ from representation to the output spaces. Figure 3 shows the overall architecture of `AIRL`; the technical details of each component are presented in Appendix B. The learning and inference processes of `AIRL` are formally stated as follows.

## 5.1 LEARNING

Non-stationary mechanisms $F^*$ and $G^*$, and classifiers $H^*$ in `AIRL` can be learned by solving an optimization problem over $T$ source domains $\{D_t\}_{t=1}^T$:

$$F^*, G^*, H^* = \arg\min_{F,G,H} \sum_{t=1}^T \mathcal{L}_{cls}^t + \alpha \mathcal{L}_{inv}^t \qquad (1)$$

where $\mathcal{L}_{cls}^t$ is the prediction loss on source domains $D_t$ and $D_{t+1}$; $\mathcal{L}_{inv}^t$ enforces the representations are invariant across a pair of consecutive domains $D_t, D_{t+1}$; hyper-parameter $\alpha$ controls the trade-off between two objectives. We note

that enforcing pairwise invariance as in objective (1) does not imply global invariance (i.e., representations that are invariant across all domains). It is because we use distinct mappings for different pairs of domains. In particular, $D_{t-1}$ and $D_t$ are aligned by two mappings $f_{t-1}$ and $g_{t-1}$ while $D_t$ and $D_{t+1}$ are aligned by two mappings $f_t$ and $g_t$.

Next, we present the detailed architecture of the *representation network* and the *classification network*. In practice, the representation mappings are often complex (e.g., ResNet [He et al., 2016] for image data, Transformer [Vaswani et al., 2017] for text data), then explicitly capturing the evolving of these mappings is challenging. We surpass this bottleneck by capturing the evolving of representation space induced by these mappings instead. Formally, our *representation network* consists of an encoder Enc which maps from input to representation spaces, and Transformer layer Trans which learns the non-stationary pattern from a sequence of source domains using attention mechanism. Given the batch sample $\mathcal{B} := \{x_t, y_t\}_{t \leq T}$ from $T$ source domains where $\{x_t, y_t\} = \{x_t^j, y_t^j\}_{j=1}^n$ are samples for domain $D_t$, the encoder first maps each input $x_t^j$ to a representation $z_t^j = \text{Enc}\left(x_t^j\right), \forall t \leq T, j \leq n$. Then, Transformer layer Trans is used to generate representation $\widehat{z}_t^j$ from the sequence $z_{\leq t}^j = \left[z_1^j, z_2^j, \cdots, z_t^j\right]$. Specifically, $\forall j, t$, Trans leverages four feed-forward networks $Q, K, V, U$ to compute $\widehat{z}_t^j$ as follow:

$$\widehat{z}_t^j = \left(a_{\leq t}^j\right)^\top V\left(z_{\leq t}^j\right) + U\left(z_t^j\right)$$

$$\text{with} \quad a_{\leq t}^j = \frac{K\left(z_{\leq t}^j\right)^\top Q\left(z_t^j\right)}{\sqrt{d}} \quad (2)$$

It is worth pointing out that we do not assume data are aligned across domain sequence. In particular, due to randomness in data loading, there is no alignment between the $j^{th}$ sample in domain $D_{t-1}$ and the $j^{th}$ sample in domain $D_t$. In our design, the computational paths from $x_{t+1}^j$ to $z_{t+1}^j$ and from $x_t^j$ to $\widehat{z}_t^j$ are considered as $g_t$ and $f_t$, respectively. In particular, $z_{t+1}^j = g_t(x_{t+1}^j)$ and $\widehat{z}_t^j = f_t(x_t^j)$. The main goal of this design is as follows: By incorporating historical data into computation, representation space constructed by $f_t$ can capture evolving pattern across domain sequence. However, this design requires access to the historical data during inference which might not be feasible in practice. To avoid it, we enforce $g_t$, which obviates the need to access historical data, to mimic representation space constructed by $f_t$.

As shown in Remark 1, our goal is to enforce invariant representation constraint (i.e., $\mathcal{L}_{inv}^t$ in objective (1)) for every pair of two consecutive domains $D_t, D_{t+1}$ constructed by $f_t$ and $g_t$ instead of learning a network that achieves invariant representations for all source domains together. Thus, the representations constructed by $f_t$ and $g_t$ might not be aligned

with the ones constructed by $f_{t'}$ and $g_{t'}$. After mapping data to the representation spaces, the *classification network* are used to generate the classifier sequence $H$. Due to the simplicity of the classifier in practice (i.e., 1 or 2-layer network), we leverage long short-term memory [Hochreiter and Schmidhuber, 1997] LSTM to explicitly capture the evolving of the classifier over domain sequence. Specifically, the weights of previous classifiers $h_{<t} = [h_1, h_2, \cdots, h_{t-1}]$ are vectorized and put into LSTM to generate the weights of $h_t$.

Because $f_t, g_t, h_t \,\forall t < T$ are functions of the *representation network* and *the classification network*, the weights of these two networks are updated using the backpropagated gradients for objective (1). The pseudo-code of the complete learning process for AIRL is shown in Algorithm 1. Next, we present the details of each loss term used in optimization.

**Prediction loss $\mathcal{L}_{cls}^t$:** We adopt cross-entropy loss for classification tasks. Specifically, $\mathcal{L}_{cls}^t$ for the optimization over domains $D_t, D_{t+1}$ is defined as follows.

$$\mathcal{L}_{cls}^t = \mathbb{E}_{D_t^W}\left[-\log\left(\frac{h_t(f_t(X))_Y}{\sum_{y' \in \mathcal{Y}} h_t(f_t(X))_{y'}}\right)\right]$$
$$+ \mathbb{E}_{D_{t+1}}\left[-\log\left(\frac{h_t(g_t(X))_Y}{\sum_{y' \in \mathcal{Y}} h_t(g_t(X))_{y'}}\right)\right] \quad (3)$$

**Invariant representation constraint $\mathcal{L}_{inv}^t$:** It aims to minimize the distance between $f_{t\sharp} P_{D_t^W}^{Z|Y=y}$ and $g_{t\sharp} P_{D_{t+1}}^{Z|Y=y}$, $\forall y \in \mathcal{Y}$, two conditional distributions induced from domains $D_t^W$ and $D_{t+1}$ using representation mappings $f_t, g_t$, respectively. In other words, for any inputs $X$ from domains $D_t^W$ and $X'$ from $D_{t+1}$ whose labels are the same, we need to find representation mappings $f_t, g_t$ such that the representations $f_t(X), g_t(X')$ have similar distributions. Inspired by correlation alignment loss [Sun and Saenko, 2016], we enforce this constraint by using the following as loss $\mathcal{L}_{inv}^t$:

$$\mathcal{L}_{inv}^t = \sum_{y \in \mathcal{Y}} \frac{1}{4d^2} \left\|C_t^y - C_{t+1}^y\right\|_F^2 \quad (4)$$

where $d$ is the dimension of representation space $\mathcal{Z}$, $\|\cdot\|_F^2$ is the squared matrix Frobenius norm, and $C_t^y$ and $C_{t+1}^y$ are covariance matrices defined as follows:

$$C_t^y = \frac{1}{n_y^t - 1}\left(f_t\left(\mathbf{X}_t^y\right)^\top f_t\left(\mathbf{X}_t^y\right)\right.$$
$$\left. -\frac{1}{n_y^t}\left(\mathbf{1}^\top f_t\left(\mathbf{X}_t^y\right)\right)^\top \left(\mathbf{1}^\top f_t\left(\mathbf{X}_t^y\right)\right)\right) \quad (5)$$
$$C_{t+1}^y = \frac{1}{n_y^{t+1} - 1}\left(g_t\left(\mathbf{X}_{t+1}^y\right)^\top g_t\left(\mathbf{X}_{t+1}^y\right)\right.$$
$$\left. -\frac{1}{n_y^{t+1}}\left(\mathbf{1}^\top g_t\left(\mathbf{X}_{t+1}^y\right)\right)^\top \left(\mathbf{1}^\top g_t\left(\mathbf{X}_{t+1}^y\right)\right)\right) \quad (6)$$

where $\mathbf{1}$ is the column vector with all elements equal to 1, $\mathbf{X}_t^y = \{x_i : x_i \in D_t^W, y_i = y\}$ is the matrix whose

**Algorithm 1:** Learning process for `AIRL`

---
**Input:** Training datasets from $T$ source domains $\{D_t\}_{t=1}^{T}$, *representation network* = {Enc, Trans}, *classification network* = {LSTM, $h_1$}, $\alpha, n$

**Output:** Trained Enc, Trans, LSTM, $h_1^*$

1   $L_{inv} = 0, L_{cls} = 0$
   /* Estimate $\{w_y^t\}_{y \in \mathcal{Y}, t < T}$ for important weighting      */
2   **for** $t = 1 : T - 1$ **do**
3     **for** $y \in \mathcal{Y}$ **do**
4       $w_y^t = P_{D_{t+1}}^{Y=y} / P_{D_t}^{Y=y}$
    /* Learn weights for Enc, Trans, LSTM */
5   **while** *learning is not end* **do**
6     Sample batch $\mathcal{B} = \{x_t, y_t\}_{t=1}^{T} \sim \{D_t\}_{t=1}^{T}$ where $\{x_t, y_t\} = \left\{x_t^j, y_t^j\right\}_{j=1}^{n}$
7     $z_1 = \text{Enc}(x_1)$
8     **for** $t = 1 : T - 1$ **do**
9       $z_{t+1} = \text{Enc}(x_{t+1})$
10      $\widehat{z}_t = \text{Trans}(z_{\leq t})$
11      $\{\widehat{z}_t(w), y_t(w)\} = \text{Reweight } \{\widehat{z}_t, y_t\}$ with $w^t = \{w_y^t\}_{y \in \mathcal{Y}}$
12      Calculate $L_{inv}^t$ from $\widehat{z}_t(w), z_{t+1}$ by Eq. (4)
13      $L_{inv} = L_{inv} + L_{inv}^t$
14      **if** $t > 1$ **then**
15        $h_t = \text{LSTM}(h_{<t})$
16      Calculate $L_{cls}^t$ from $y_t(w), y_{t+1}, h_t(\widehat{z}_t(w)), h_t(z_{t+1})$ by Eq. (3)
17      $L_{cls} = L_{cls} + L_{cls}^t$
18     Update Enc, Trans, LSTM, $\widehat{h}_1$ by optimizing $L_{inv} + \alpha L_{cls}$

---

columns are $\{x_i\}$, $f_t$ and $g_t$ are column-wise operations applied to $\mathbf{X}_t^y$ and $\mathbf{X}_{t+1}^y$, respectively, and $n_y^t$ is cardinality of $\mathbf{X}_t^y$.

## 5.2   INFERENCE

At the inference stage, the well-trained *representation network* and *classification network* can be used to make predictions about input $x$ from target domain sequence $\{D_t\}_{t=T+1}^{T+K}$. In particular, we first map input $x$ in domain $D_t$ to representation $z$ using the encoder Enc in the *representation network* (i.e., $g_{t-1}^*$). Then the *classification network* (i.e., LSTM) is used sequentially to generate $h_{t-1}^*$ from the sequence of classifiers $[h_1^*, \cdots, h_{t-2}^*]$, and the prediction about $z$ can be made by $h_{t-1}^*$. Note that at the learning stage, both $g_{t-1}^*$ and $f_t^*$ are used to map input $x$ from domain $D_t$ to the representation space while at the inference stage, only $g_{t-1}^*$ is needed for target domain $D_t$ (we do not use $f_t^*$ because it requires access to data from all domains $\{D_{t'}\}_{t' \leq t}$

which generally are not available during inference). The complete inference process is shown in Algorithm 2.

---
**Algorithm 2:** Inference process for `AIRL`

---
**Input:** Testing dataset from target domain $D_t (t \in \{T + 1, \cdots, T + K\})$, trained Enc, LSTM, $h_1^*$

**Output:** Predictions for testing dataset

1   [t]
2   **for** $t' = 2 : (t - 1)$ **do**
3     $h_{t'}^* = \text{LSTM}\left(h_{<t'}^*\right)$
4   **while** *inference is not end* **do**
5     Sample batch $\mathcal{B} = x_t \sim D_t$
6     $z_t = \text{Enc}(x_t)$
7     Generate predictions $h_{t-1}^*(z_t)$

---

## 6   EXPERIMENTS

In this section, we present the experimental results of the proposed `AIRL` and compare `AIRL` with a wide range of existing algorithms. We evaluate these algorithms on synthetic and real-world datasets. Next, we first introduce the experimental setup and then present the empirical results.

**Experimental setup.** Datasets and baselines used in the experiments are briefly introduced below. Their details are shown in Appendix C.

*Datasets.* We consider five datasets: **Circle** [Pesaranghader and Viktor, 2016] (a synthetic dataset containing 30 domains where each instance is sampled from 30 two-dimensional Gaussian distributions), **Circle-Hard** (a synthetic dataset adapted from **Circle** dataset such that domains do not uniformly evolve), **RMNIST** (a semi-synthetic dataset constructed from MNIST [LeCun et al., 1998] by $R$-degree counterclockwise rotation), **Yearbook** [Ginosar et al., 2015] (a real dataset consisting of frontal-facing American high school yearbook photos from 1930-2013), and **CLEAR** [Lin et al., 2021] (a real dataset capturing the natural temporal evolution of visual concepts that spans a decade).

*Baselines.* We compare the proposed `AIRL` with existing methods from related areas, including the followings: empirical risk minimization (`ERM`), last domain training (`LD`), fine tuning (`FT`), domain invariant representation learning (`G2DM` [Albuquerque et al., 2019], `DANN` [Ganin et al., 2016], `CDANN` [Li et al., 2018b], `CORAL` [Sun and Saenko, 2016], `IRM` [Arjovsky et al., 2019]), data augmentation (`MIXUP` [Zhang et al., 2018]), continual learning (`EWC` [Kirkpatrick et al., 2017]), continuous DA (`CIDA` [Wang et al., 2020]), distributionally robust optimization (`GroupDRO` [Sagawa et al., 2019]), gradient-based DG (`Fish` [Shi et al., 2022]) contrastive learning-based DG (i.e., `SelfReg` [Kim et al., 2021]), non-stationary DG (`DRAIN` [Bai et al., 2022], `TKNets` [Zeng et al., 2024b], `LSSAE`

Table 2: Prediction performances (i.e., $\text{OOD}_{\text{Avg}}$ and $\text{OOD}_{\text{Wrt}}$) of `AIRL` and baselines under Eval-D scenario ($K = 5$). We report average results (w. standard deviation) over 5 random seeds. For **CLEAR** dataset, due to only one split between train and test sets, $\text{OOD}_{\text{Avg}}$ and $\text{OOD}_{\text{Wrt}}$ are similar.

| Algorithm | Circle | | Circle-Hard | | RMNIST | | Yearbook | | CLEAR |
| | $\text{OOD}_{\text{Avg}}$ | $\text{OOD}_{\text{Wrt}}$ | $\text{OOD}_{\text{Avg}}$ | $\text{OOD}_{\text{Wrt}}$ | $\text{OOD}_{\text{Avg}}$ | $\text{OOD}_{\text{Wrt}}$ | $\text{OOD}_{\text{Avg}}$ | $\text{OOD}_{\text{Wrt}}$ | $\text{OOD}_{\text{Avg}}$ / $\text{OOD}_{\text{Wrt}}$ |
|---|---|---|---|---|---|---|---|---|---|
| ERM | 89.63 (0.89) | 79.84 (1.84) | 66.94 (1.69) | 58.43 (0.05) | 56.61 (1.83) | 51.85 (4.15) | 90.79 (0.16) | 71.03 (1.74) | 69.04 (0.18) |
| LD | 76.60 (6.45) | 56.88 (3.74) | 58.13 (1.67) | 51.58 (1.87) | 37.54 (2.77) | 25.80 (4.12) | 77.10 (0.30) | 57.97 (0.88) | 57.01 (2.15) |
| FT | 85.57 (1.82) | 71.99 (4.11) | 59.02 (5.20) | 50.80 (2.79) | 60.73 (0.87) | 47.30 (3.77) | 87.04 (0.58) | 66.83 (2.22) | 66.71 (0.46) |
| DANN | 88.80 (1.17) | 78.32 (3.23) | 65.10 (0.93) | 56.68 (0.59) | 58.25 (1.15) | 53.61 (1.61) | 90.57 (0.22) | 69.58 (1.38) | 67.48 (1.19) |
| CDANN | 89.75 (0.14) | 80.75 (2.97) | 64.05 (1.33) | 58.68 (0.22) | 58.19 (0.93) | 54.45 (1.40) | 90.46 (0.30) | 70.37 (1.44) | 66.12 (0.37) |
| G2DM | 89.40 (2.27) | 79.61 (2.94) | 67.75 (2.69) | 59.65 (1.61) | 57.62 (0.39) | 53.93 (0.31) | 87.57 (0.37) | 66.69 (1.15) | 56.98 (2.77) |
| CORAL | 90.13 (0.52) | 83.14 (1.27) | 66.12 (1.48) | 59.62 (1.17) | 51.41 (2.63) | 44.95 (3.64) | 90.41 (0.20) | 69.53 (2.00) | 70.96 (1.06) |
| GROUPDRO | 90.50 (1.75) | 81.07 (6.12) | 67.08 (1.67) | 58.51 (0.12) | 54.37 (2.98) | 46.21 (5.69) | 90.65 (0.20) | 71.21 (1.51) | 70.63 (0.04) |
| MIXUP | 88.49 (0.86) | 76.78 (2.49) | 63.03 (1.53) | 56.21 (1.20) | 52.13 (2.54) | 34.60 (16.81) | 89.75 (0.05) | 68.73 (1.36) | 69.58 (0.99) |
| IRM | 85.78 (1.11) | 74.80 (1.73) | 62.43 (2.70) | 54.96 (1.78) | 26.96 (1.11) | 16.25 (1.87) | 84.65 (0.31) | 64.30 (2.44) | 49.54 (1.08) |
| SELFREG | 90.33 (0.14) | 82.20 (0.93) | 68.23 (2.47) | 60.28 (0.90) | 50.58 (2.35) | 42.15 (4.63) | 91.47 (0.12) | 73.88 (0.37) | 69.18 (0.68) |
| FISH | 90.65 (0.25) | 79.09 (2.46) | 62.69 (0.63) | 56.97 (0.49) | 56.53 (1.32) | 52.23 (1.47) | 89.92 (0.20) | 70.58 (0.90) | 69.46 (0.47) |
| EWC | 89.18 (1.72) | 79.59 (4.63) | 68.31 (3.31) | 61.34 (2.18) | 66.53 (1.26) | 50.63 (5.35) | 89.47 (0.17) | 59.09 (7.70) | 45.58 (4.92) |
| CIDA | 87.25 (0.88) | 77.91 (0.23) | 65.38 (2.77) | 58.15 (0.88) | 53.42 (4.35) | 35.21 (17.85) | 91.29 (0.16) | 70.19 (1.45) | 65.10 (0.12) |
| DRAIN | 86.78 (0.65) | 74.57 (1.82) | 67.44 (4.65) | 57.76 (3.42) | 67.09 (4.06) | 59.49 (8.31) | 89.62 (0.39) | 70.36 (2.32) | 64.67 (0.65) |
| TKNets | 91.76 (0.16) | **83.35 (1.32)** | 64.19 (0.95) | 59.94 (0.18) | 74.39 (0.23) | 71.03 (0.37) | 92.11 (0.26) | 75.04 (1.16) | 64.05 (0.64) |
| LSSAE[1] | 90.21 (1.95) | 80.92 (3.53) | 66.43 (0.81) | 61.22 (0.71) | 33.30 (2.14) | 18.83 (3.85) | 60.48 (4.99) | 50.35 (4.67) | 22.61 (0.25) |
| DDA | 72.06 (4.51) | 48.81 (0.97) | 65.26 (3.20) | 56.16 (2.45) | **78.18** (0.88) | 73.70 (0.31) | 86.72 (0.56) | 67.60 (2.66) | 70.12 (1.10) |
| AIRL | **92.28 (0.27)** | 82.81 (2.70) | **73.50 (2.21)** | **63.29 (1.26)** | 77.49 (0.86) | **74.99 (0.57)** | **93.10 (0.21)** | **78.22 (0.92)** | **73.04** (0.67) |

Table 3: Ablation study for `AIRL` on **Circle-Hard** dataset under Eval-D scenario ($K = 5$).

| LSTM | Trans | $\mathcal{L}_{inv}$ | $\text{OOD}_{\text{Avg}}$ | $\text{OOD}_{\text{Wrt}}$ |
|---|---|---|---|---|
| ✗ | ✓ | ✓ | 69.06 | 61.05 |
| ✓ | ✗ | ✓ | 65.51 | 58.69 |
| ✓ | ✗ | ✗ | 68.33 | 60.16 |
| ✓ | ✓ | ✓ | **73.50** | **63.29** |

[Qin et al., 2022], and `DDA` [Zeng et al., 2023]). To ensure a fair comparison, we adopt similar architectures for `AIRL` and baselines, including both representation mapping and classifier. The implementation details are in Appendix B.

*Evaluation method.* In the experiments, models are trained on a sequence of source domains $\mathcal{D}_{src}$, and their performance is evaluated on target domains $\mathcal{D}_{tgt}$ under two different scenarios: Eval-S and Eval-D. In the scenario Eval-S, models are trained one time on the first half of domain sequence $\mathcal{D}_{src} = [D_1, D_2, \cdots, D_T]$ and are then deployed to make predictions on the second half of domain sequence $\mathcal{D}_{tgt} = [D_{T+1}, D_{T+2}, \cdots, D_{2T}](K = T)$. In the scenario Eval-D, source and target domains are not static but are updated periodically as new data/domain becomes available. For each of these two scenarios, we use two accuracy measures, $\text{OOD}_{\text{Avg}}$ and $\text{OOD}_{\text{Wrt}}$, to evaluate the average- and worst-case performances. Their details are shown in Appendix C. We train each model with 5 different random seeds and report the average prediction performances.

**Results.** Next, we evaluate the model performance under Eval-D scenario (Results for Eval-S are in Appendix C).

*Non-stationary DG results.* Performance of `AIRL` and baselines on synthetic (i.e., **Circle**, **Circle-Hard**) and real-world (i.e., **RMNIST**, **Yearbook**) data are presented in Table 2. We observe that `AIRL` consistently outperforms other methods over all datasets and metrics. These results indicate that `AIRL` can effectively capture non-stationary patterns across domains, and such patterns can be leveraged to learn the models that generalize better on target domains compared to the baselines. Among baselines, methods designed specifically for non-stationary DG (i.e., `DRAIN`, `DPNET`, `LSSAE`) and continual learning method (i.e., `EWC`) achieve better performance than other methods. However, such improvement is inconsistent across datasets.

*Comparison with non-stationary DG methods.* `DPNET` assumes that the evolving pattern between two consecutive domains is constant and the distances between them are small. Thus, this method does not achieve good performance for **Circle-Hard** dataset where distance between two consecutive domains is proportional to domain index. `DRAIN` utilizes Bayesian framework and generates the whole models at every domain. This method, however, is only capable for small neural networks and does not scale well to real-world applications. Moreover, `DPNET`, `DRAIN`, and `DDA` can only generalize to a single subsequent target domain. `LSSAE` leverages sequential variational auto-encoder [Li and Mandt, 2018] to learn non-stationary pattern. However, this model assumes the availability of aligned data across domain sequence, which may pose challenges to its per-

---

[1]We have observed that the training process of `LSSAE` with our image encoders on image datasets fails to converge.

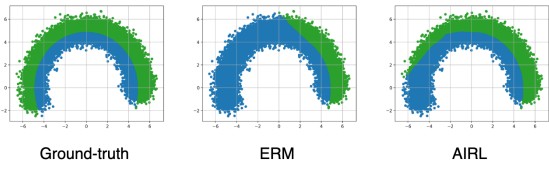

Figure 4: Visualization of predictions on **Circle-Hard** dataset generated by `ERM` and `AIRL`.

formance in non-stationary DG. In contrast, `AIRL` is not limited to the constantly evolving pattern. It is also scalable to large neural networks and can handle multiple target domains. In particular, compared to the base model (`ERM`), our method has only one extra Transformer and LSTM layers. Note that these layers are used during training only. In the inference stage, predictions are made by Enc and classifier pre-generated by LSTM which then results in a similar inference time with `ERM`.

*Decision boundary visualization.* We conduct a quantitative analysis for our method by visualizing its predictions on **Circle-Hard** dataset. We train models (i.e., `AIRL` and `ERM`) on the first 10 domains (right half) and evaluate on the remaining 10 domains (left half). As depicted in Figure 4, our method, designed to capture non-stationary patterns across domains, generates more accurate predictions for target domains compared to ERM.

*Ablation studies.* We conduct experiments to investigate the roles of each component in `AIRL`. In particular, we compare `AIRL` with its variants; each variant is constructed by removing LSTM (i.e., use fixed classifier instead), Trans (i.e., use fixed representation instead), $L_{inv}$ (i.e., without invariant constraint) from the model. As shown in Table 3, model performance deteriorates when removing any of them. These results validate our theorems and demonstrate the effectiveness of each component.

*Limitations.* While `AIRL` consistently outperforms existing methods across all datasets and metrics, we acknowledge certain limitations in our work. Regarding theoretical analysis, we presently lack an effective method to estimate non-stationary complexity from finite data. Concerning algorithm design, our method is unable to address scenarios where data from all source domains are not simultaneously available during training (i.e., online learning). Moreover, it may not be generalized to every non-stationary environment in some specific cases. This is due to the reliance of our method on the selection of hypothesis classes $\mathcal{F}, \mathcal{G}$.

## 7 CONCLUSION

In this paper, we theoretically and empirically studied domain generalization under non-stationary environments. We first established the upper bounds of prediction error on target domains, and then proposed a representation learning-based method that learns adaptive invariant representations across source domains. The resulting models trained with the proposed method can generalize well to unseen target domains. Experiments on both synthetic and real data demonstrate the effectiveness of our proposed method.

## Acknowledgements

This work was funded in part by the National Science Foundation under award number IIS-2145625, by the National Institutes of Health under award number R01GM141279, and by The Ohio State University President's Research Excellence Accelerator Grant.

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

# Non-stationary Domain Generalization: Theory and Algorithm (Supplementary Material)

**Thai-Hoang Pham**[1,2]       **Xueru Zhang**[1]       **Ping Zhang**[1,2]

[1]Department of Computer Science and Engineering, The Ohio State University, USA
[2]Department of Biomedical Informatics, The Ohio State University, USA
{pham.375,zhang.12807,zhang.10631}@osu.edu

## A  PROOFS

### A.1  ADDITIONAL LEMMAS

**Lemma 1.** *Given two domains $D_t$ and $D_{t'}$, then for any classifier $h \in \mathcal{H}$, the expected error of $h$ in domain $D_t$ can be upper bounded:*

$$\epsilon_{D_t}(h) \leq \epsilon_{D_{t'}}(h) + \sqrt{2}C \times \mathcal{D}_{JS}\left(P_{D_t}^{X,Y} \parallel P_{D_{t'}}^{X,Y}\right)^{1/2}$$

*where $\mathcal{D}_{JS}(\cdot \parallel \cdot)$ is JS-divergence between two distributions.*

**Proof of Lemma 1**   Let $D_{KL}(\cdot \parallel \cdot)$ be KL-divergence and $U = (X, Y)$ and $L(U) = L(h(X), Y)$. We first prove $\int_{\mathcal{E}} \left| P_{D_t}^{U=u} - P_{D_{t'}}^{U=u} \right| du = \frac{1}{2} \int \left| P_{D_t}^{U=u} - P_{D_{t'}}^{U=u} \right| du$ where $\mathcal{E}$ is the event that $P_{D_t}^{U=u} \geq P_{D_{t'}}^{U=u}$ $(*)$ as follows:

$$
\begin{aligned}
\int_{\mathcal{E}} \left| P_{D_t}^{U=u} - P_{D_{t'}}^{U=u} \right| du &= \int_{\mathcal{E}} \left( P_{D_t}^{U=u} - P_{D_{t'}}^{U=u} \right) du \\
&= \int_{\mathcal{E} \cup \overline{\mathcal{E}}} \left( P_{D_t}^{U=u} - P_{D_{t'}}^{U=u} \right) du - \int_{\overline{\mathcal{E}}} \left( P_{D_t}^{U=u} - P_{D_{t'}}^{U=u} \right) du \\
&\overset{(1)}{=} \int_{\overline{\mathcal{E}}} \left( P_{D_{t'}}^{U=u} - P_{D_t}^{U=u} \right) du \\
&= \int_{\overline{\mathcal{E}}} \left| P_{D_t}^{U=u} - P_{D_{t'}}^{U=u} \right| du \\
&= \frac{1}{2} \int \left| P_{D_t}^{U=u} - P_{D_{t'}}^{U=u} \right| du
\end{aligned}
$$

where $\overline{\mathcal{E}}$ is the complement of $\mathcal{E}$. We have $\overset{(1)}{=}$ because $\int_{\mathcal{E} \cup \overline{\mathcal{E}}} \left( P_{D_t}^{U=u} - P_{D_{t'}}^{U=u} \right) du = \int_{\mathcal{U}} \left( P_{D_t}^{U=u} - P_{D_{t'}}^{U=u} \right) du = 0$. Then, we have:

$$\epsilon_{D_t}(h) = \mathbb{E}_{D_t}[L(U)]$$

$$= \int_{\mathcal{U}} L(u) P_{D_t}^{U=u} du$$

$$= \int_{\mathcal{U}} L(u) P_{D_{t'}}^{U=u} du + \int_{\mathcal{U}} L(u) \left( P_{D_t}^{U=u} - P_{D_{t'}}^{U=u} \right) du$$

$$= \mathbb{E}_{D_{t'}}[L(U)] + \int_{\mathcal{U}} L(u) \left( P_{D_t}^{U=u} - P_{D_{t'}}^{U=u} \right) du$$

$$= \epsilon_{D_{t'}}(h) + \int_{\mathcal{E}} L(u) \left( P_{D_t}^{U=u} - P_{D_{t'}}^{U=u} \right) du + \int_{\overline{\mathcal{E}}} L(u) \left( P_{D_t}^{U=u} - P_{D_{t'}}^{U=u} \right) du$$

$$\overset{(2)}{\leq} \epsilon_{D_{t'}}(h) + \int_{\mathcal{E}} L(u) \left( P_{D_t}^{U=u} - P_{D_{t'}}^{U=u} \right) du$$

$$\overset{(3)}{\leq} \epsilon_{D_{t'}}(h) + C \int_{\mathcal{E}} \left( P_{D_t}^{U=u} - P_{D_{t'}}^{U=u} \right) du$$

$$= \epsilon_{D_{t'}}(h) + C \int_{\mathcal{E}} \left| P_{D_t}^{U=u} - P_{D_{t'}}^{U=u} \right| du$$

$$\overset{(4)}{=} \epsilon_{D_{t'}}(h) + \frac{C}{2} \int \left| P_{D_t}^{U=u} - P_{D_{t'}}^{U=u} \right| du$$

$$\overset{(5)}{\leq} \epsilon_{D_{t'}}(h) + \frac{C}{2} \sqrt{2 \min \left( \mathcal{D}_{KL}\left( P_{D_{t'}}^{U} \parallel P_{D_t}^{U} \right), \mathcal{D}_{KL}\left( P_{D_t}^{U} \parallel P_{D_{t'}}^{U} \right) \right)}$$

$$\leq \epsilon_{D_{t'}}(h) + \frac{C}{\sqrt{2}} \sqrt{\mathcal{D}_{KL}\left( P_{D_{t'}}^{U} \parallel P_{D_t}^{U} \right)} \quad (**)$$

We have $\overset{(2)}{\leq}$ because $\int_{\overline{\mathcal{E}}} L(u) \left( P_{D_t}^{U=u} - P_{D_{t'}}^{U=u} \right) du \leq 0$; $\overset{(3)}{\leq}$ because $L(u)$ is non-negative function and is bounded by $C$; $\overset{(4)}{=}$ by using $(*)$; $\overset{(5)}{\leq}$ by using Pinsker's inequality between total variation norm and KL-divergence.

Let $P_{D_{t,t'}}^{U} = \frac{1}{2}\left( P_{D_t}^{U} + P_{D_{t'}}^{U} \right)$. Apply $(**)$ for two domains $D_t$ and $D_{t,t'}$, we have:

$$\epsilon_{D_t}(h) \leq \epsilon_{D_{t,t'}}(h) + \frac{C}{\sqrt{2}} \sqrt{\mathcal{D}_{KL}\left( P_{D_t}^{U} \parallel P_{D_{t,t'}}^{U} \right)} \tag{7}$$

Apply $(**)$ again for two domains $D_{t,t'}$ and $D_{t'}$, we have:

$$\epsilon_{D_{t,t'}}(h) \leq \epsilon_{D_{t'}}(h) + \frac{C}{\sqrt{2}} \sqrt{\mathcal{D}_{KL}\left( P_{D_{t'}}^{U} \parallel P_{D_{t,t'}}^{U} \right)} \tag{8}$$

Adding Eq. (7) to Eq. (8) and subtracting $\epsilon_{D_{t,t'}}$, we have:

$$\epsilon_{D_t}(h) \leq \epsilon_{D_{t'}}(h) + \frac{C}{\sqrt{2}} \left( \sqrt{\mathcal{D}_{KL}\left( P_{D_t}^{U} \parallel P_{D_{t,t'}}^{U} \right)} + \sqrt{\mathcal{D}_{KL}\left( P_{D_{t'}}^{U} \parallel P_{D_{t,t'}}^{U} \right)} \right)$$

$$\overset{(6)}{\leq} \epsilon_{D_{t'}}(h) + \frac{C}{\sqrt{2}} \sqrt{2\left( \mathcal{D}_{KL}\left( P_{D_t}^{U} \parallel P_{D_{t,t'}}^{U} \right) + \mathcal{D}_{KL}\left( P_{D_{t'}}^{U} \parallel P_{D_{t,t'}}^{U} \right) \right)}$$

$$= \epsilon_{D_{t'}}(h) + \frac{C}{\sqrt{2}} \sqrt{4 \mathcal{D}_{JS}\left( P_{D_{t'}}^{U} \parallel P_{D_t}^{U} \right)}$$

$$= \epsilon_{D_{t'}}(h) + \sqrt{2} C \sqrt{\mathcal{D}_{JS}\left( P_{D_{t'}}^{U} \parallel P_{D_t}^{U} \right)}$$

We have $\overset{(6)}{\leq}$ by using Cauchy–Schwarz inequality.

**Lemma 2.** *Given domain $D$, then for any $\delta > 0$, with probability at least $1 - \delta$ over samples $S$ of size $n$ drawn i.i.d from domain $D$, for all $h \in \mathcal{H}$, the expected error of $h$ in domain $D$ can be upper bounded:*

$$\epsilon_D(h) \leq \epsilon_S(h) + \frac{2B}{\sqrt{n}} + C\sqrt{\frac{\log(1/\delta)}{2n}}$$

**Proof of Lemma 2** We start from the Rademacher bound Koltchinskii and Panchenko [2000] which is stated as follows.

**Lemma 3.** *Rademacher Bounds. Let $\mathcal{F}$ be a family of functions mapping from $Z$ to $[0, 1]$. Then, for any $0 < \delta < 1$, with probability at least $1 - \delta$ over sample $S = \{z_1, \cdots, z_n\}$, the following holds for all $f \in \mathcal{F}$:*

$$\mathbb{E}\left[f^Z\right] \leq \frac{1}{n}\sum_{i=1}^{n} f(z_i) + 2\mathcal{R}_n(\mathcal{F}) + \sqrt{\frac{\log(1/\delta)}{2n}}$$

*where $\mathcal{R}_n(\mathcal{F})$ is a Rademacher complexity of function class $\mathcal{F}$.*

We then apply Lemma 3 to our setting with $Z = (X, Y)$, the loss function $L$ bounded by $C$, and the function class $\mathcal{L}_\mathcal{H} = \{(x, y) \to L(h(x), y) : h \in \mathcal{H}\}$. In particular, we scale the loss function $L$ to $[0, 1]$ by dividing by C and denote the new class of scaled loss functions as $\mathcal{L}_\mathcal{H}/C$. Then, for any $\delta > 0$, with probability at least $1 - \delta$, we have:

$$
\begin{aligned}
\frac{\epsilon_D(h)}{C} &\leq \frac{\epsilon_S(h)}{C} + 2\mathcal{R}_n(\mathcal{L}_\mathcal{H}/C) + \sqrt{\frac{\log(1/\delta)}{2n}} \\
&\overset{(1)}{=} \frac{\epsilon_S(h)}{C} + \frac{2}{C}\mathcal{R}_n(\mathcal{L}_\mathcal{H}) + \sqrt{\frac{\log(1/\delta)}{2n}} \\
&\overset{(2)}{\leq} \frac{\epsilon_S(h)}{C} + \frac{2B}{C\sqrt{n}} + \sqrt{\frac{\log(1/\delta)}{2n}}
\end{aligned}
\tag{9}
$$

We have $\overset{(1)}{=}$ by using the property of Redamacher complexity that $\mathcal{R}_n(\alpha\mathcal{F}) = \alpha\mathcal{R}_n(\mathcal{F})$, $\overset{(2)}{\leq}$ because of bounded Rademacher complexity assumption. We derive Lemma 2 by multiplying Eq. (9) by C.

**Lemma 4.** *Given domain sequence $\{D_t\}$, dataset sequence $\{S_t\}$ sampled from $\{D_t\}$, M-optimal model sequence $H^M = \left\{h_1^M, \cdots, h_{T+K}^M\right\}$, M-empirical optimal model sequence $\widehat{H}_M = \left\{\widehat{h}_1^M, \cdots, \widehat{h}_{T+K}^M\right\}$, then for any $t$ and any $\delta > 0$, with probability at least $1 - \delta$ over samples $S_t$ of size $n$ drawn i.i.d from domain $D_t$, we have:*

$$\epsilon_{D_t}\left(\widehat{h}_t^M\right) \leq \epsilon_{D_t}\left(h_t^M\right) + 2\sqrt{2}C\mathcal{D}_{JS}\left(D_t \parallel D_t^M\right)^{1/2} + \frac{4B}{\sqrt{n}} + \sqrt{\frac{2\log(1/\delta)}{n}}$$

**Proof of Lemma 4** We have:

$$
\begin{aligned}
\epsilon_{D_t}\left(\widehat{h}_t^M\right) &\overset{(1)}{\leq} \epsilon_{D_t^M}\left(\widehat{h}_t^M\right) + \sqrt{2}C\mathcal{D}_{JS}\left(D_t \parallel D_t^M\right)^{1/2} \\
&\overset{(2)}{\leq} \epsilon_{S_t^M}\left(\widehat{h}_t^M\right) + \sqrt{2}C\mathcal{D}_{JS}\left(D_t \parallel D_t^M\right)^{1/2} + \frac{2B}{\sqrt{n}} + \sqrt{\frac{\log(1/\delta')}{2n}} \quad \text{(w.p} \geq 1 - \delta') \\
&\overset{(3)}{\leq} \epsilon_{S_t^M}\left(h_t^M\right) + \sqrt{2}C\mathcal{D}_{JS}\left(D_t \parallel D_t^M\right)^{1/2} + \frac{2B}{\sqrt{n}} + \sqrt{\frac{\log(1/\delta')}{2n}} \\
&\overset{(4)}{\leq} \epsilon_{D_t^M}\left(h_t^M\right) + \sqrt{2}C\mathcal{D}_{JS}\left(D_t \parallel D_t^M\right)^{1/2} + \frac{4B}{\sqrt{n}} + \sqrt{\frac{2\log(1/\delta')}{n}} \quad \text{(w.p} \geq 1 - \delta') \\
&\overset{(5)}{\leq} \epsilon_{D_t}\left(h_t^M\right) + 2\sqrt{2}C\mathcal{D}_{JS}\left(D_t \parallel D_t^M\right)^{1/2} + \frac{4B}{\sqrt{n}} + \sqrt{\frac{2\log(1/\delta')}{n}}
\end{aligned}
$$

We have $\overset{(1)}{\leq}$ by using Lemma 1 for $\epsilon_{D_t}\left(\widehat{h}_t^M\right)$, $\overset{(2)}{\leq}$ by using Lemma 2 for $\epsilon_{D_t^M}\left(\widehat{h}_t^M\right)$, $\overset{(3)}{\leq}$ because $\widehat{h}_t^M = \arg\min_{h\in\mathcal{H}} \epsilon_{S_t^M}(h)$, $\overset{(4)}{\leq}$ by using Lemma 2 for $\epsilon_{S_t^M}\left(h_t^M\right)$, $\overset{(5)}{\leq}$ by using Lemma 1 for $\epsilon_{D_t^M}\left(\widehat{h}_t^M\right)$. Finally, using union bound for $\overset{(2)}{\leq}$ and $\overset{(4)}{\leq}$, and denote $\delta = 2\delta'$, we have:

$$\epsilon_{D_t}\left(\widehat{h}_t^M\right) \leq \epsilon_{D_t}\left(h_t^M\right) + 2\sqrt{2}C\mathcal{D}_{JS}\left(D_t \parallel D_t^M\right)^{1/2} + \frac{4B}{\sqrt{n}} + \sqrt{\frac{2\log(1/\delta)}{n}}$$

**Lemma 5.** *Given domain sequence $\{D_t\}$, dataset sequence $\{S_t\}$ sampled from $\{D_t\}$, $M$-optimal model sequence $H^M = \{h_1^M, \cdots, h_{T+K}^M\}$, $M$-empirical optimal model sequence $\widehat{H}_M = \{\widehat{h}_1^M, \cdots, \widehat{h}_{T+K}^M\}$, then for any $t$, we have:*

$$\epsilon_{D_t}\left(h_t^M\right) \leq \epsilon_{D_t}\left(\widehat{h}_t^M\right) + 2\sqrt{2}C\mathcal{D}_{JS}\left(D_t \parallel D_t^M\right)^{1/2}$$

**Proof of Lemma 5**   We have:

$$
\begin{aligned}
\epsilon_{D_t}\left(h_t^M\right) &\overset{(1)}{\leq} \epsilon_{D_t^M}\left(h_t^M\right) + \sqrt{2}C\mathcal{D}_{JS}\left(D_t \parallel D_t^M\right)^{1/2} \\
&\overset{(2)}{\leq} \epsilon_{D_t^M}\left(\widehat{h}_t^M\right) + \sqrt{2}C\mathcal{D}_{JS}\left(D_t \parallel D_t^M\right)^{1/2} \\
&\overset{(3)}{\leq} \epsilon_{D_t}\left(\widehat{h}_t^M\right) + 2\sqrt{2}C\mathcal{D}_{JS}\left(D_t \parallel D_t^M\right)^{1/2}
\end{aligned}
$$

We have $\overset{(1)}{\leq}$ by using Lemma 1 for $\epsilon_{D_t}\left(h_t^M\right)$, $\overset{(2)}{\leq}$ because $h_t^M = \arg\min_{h \in \mathcal{H}} \epsilon_{D_t^M}(h)$, $\overset{(3)}{\leq}$ by using Lemma 1 for $\epsilon_{D_t^M}\left(\widehat{h}_t^M\right)$.

## A.2   PROOF OF MAIN THEOREMS.

### A.2.1   Proof of Theorem 1

We have:

$$
\begin{aligned}
E_{tgt}\left(\widehat{H}^M\right) &= \frac{1}{K}\sum_{t=T+1}^{T+K} \epsilon_{D_t}\left(\widehat{h}_t^M\right) \\
&\overset{(1)}{\leq} \frac{1}{K}\sum_{t=T+1}^{T+K}\left(\epsilon_{D_t}\left(h_t^M\right) + 2\sqrt{2}C \times \mathcal{D}_{JS}\left(D_t \parallel D_t^M\right)^{1/2} + \frac{4B}{\sqrt{n}} + \sqrt{\frac{2\log(1/\delta')}{n}}\right) \quad \text{(w.p} \geq 1 - \delta') \\
&= E_{tgt}\left(H^M\right) + 2\sqrt{2}C \times D_{tgt}\left(M\right) + \frac{4B}{\sqrt{n}} + \sqrt{\frac{2\log(1/\delta')}{n}} \\
&= E_{src}\left(H^M\right) + \left(E_{tgt}\left(H^M\right) - E_{src}\left(H^M\right)\right) + 2\sqrt{2}C \times \left(D_{src}\left(M\right) + \left(D_{tgt}\left(M\right) - D_{src}\left(M\right)\right)\right) \\
&\quad + \frac{4B}{\sqrt{n}} + \sqrt{\frac{2\log(1/\delta')}{n}} \\
&\overset{(2)}{\leq} \frac{1}{T}\sum_{t=1}^{T}\epsilon_{D_t}\left(h_t^M\right) + \Phi(\mathcal{M}) + 2\sqrt{2}C \times D_{src}\left(M\right) + 2\sqrt{2}C \times \Phi(\mathcal{M}, \mathcal{H}) + \frac{4B}{\sqrt{n}} + \sqrt{\frac{2\log(1/\delta')}{n}} \\
&\overset{(3)}{\leq} \frac{1}{T}\sum_{t=1}^{T}\left(\epsilon_{D_t}\left(\widehat{h}_t^M\right) + 2\sqrt{2}C \times \mathcal{D}_{JS}\left(D_t \parallel D_t^M\right)^{1/2}\right) + \Phi(\mathcal{M}) + 2\sqrt{2}C \times D_{src}\left(M\right) + 2\sqrt{2}C \times \Phi(\mathcal{M}, \mathcal{H}) \\
&\quad + \frac{4B}{\sqrt{n}} + \sqrt{\frac{2\log(1/\delta')}{n}} \\
&\overset{(4)}{\leq} \frac{1}{T}\sum_{t=1}^{T}\epsilon_{D_t^M}\left(\widehat{h}_t^M\right) + 5\sqrt{2}C \times D_{src}\left(M\right) + \Phi(\mathcal{M}) + 2\sqrt{2}C \times \Phi(\mathcal{M}, \mathcal{H}) + \frac{4B}{\sqrt{n}} + \sqrt{\frac{2\log(1/\delta')}{n}} \\
&\overset{(5)}{\leq} \frac{1}{T}\sum_{t=1}^{T}\left(\epsilon_{S_t^M}\left(\widehat{h}_t^M\right) + \frac{2B}{\sqrt{n}} + \sqrt{\frac{\log(1/\delta')}{2n}}\right) + 5\sqrt{2}C \times D_{src}\left(M\right) + \Phi(\mathcal{M}) + 2\sqrt{2}C \times \Phi(\mathcal{M}, \mathcal{H}) \\
&\quad + \frac{4B}{\sqrt{n}} + \sum_{t=T+1}^{T+K}\sqrt{\frac{2\log(1/\delta')}{n}} \quad \text{(w.p} \geq 1 - \delta') \\
&= \widehat{E}_{src}^M\left(\widehat{H}^M\right) + 5\sqrt{2}C \times D_{src}\left(M\right) + \Phi(\mathcal{M}) + 2\sqrt{2}C \times \Phi(\mathcal{M}, \mathcal{H}) + \frac{6B}{\sqrt{n}} + 3\sqrt{\frac{\log(1/\delta')}{2n}}
\end{aligned}
$$

We have $\overset{(1)}{\le}$ by using Lemma 4 for $\epsilon_{D_t}\left(\widehat{h}_t^M\right)$, $\overset{(2)}{\le}$ because $\Phi\left(\mathcal{M},\mathcal{H}\right) = \sup\limits_{M'\in\mathcal{M}}\left(E_{tgt}\left(H^{M'}\right) - E_{src}\left(H^{M'}\right)\right)$ amd $\Phi\left(\mathcal{M}\right) = \sup\limits_{M'\in\mathcal{M}}\left(D_{tgt}\left(M'\right) - D_{src}\left(M'\right)\right)$, $\overset{(3)}{\le}$ by using Lemma 5 for $\epsilon_{D_t}\left(h_t^M\right)$, $\overset{(4)}{\le}$ by using Lemma 2 for $\epsilon_{D_t}\left(\widehat{h}_t^M\right)$, $\overset{(5)}{\le}$ by using Lemma 1 for $\epsilon_{D_t^M}\left(\widehat{h}_t^M\right)$. Finally, using union bound for $\overset{(2)}{\le}$ and $\overset{(4)}{\le}$, and denote $\delta = (T+K)\delta'$, we have:

$$E_{tgt}\left(\widehat{H}^M\right) \le \widehat{E}_{src}^M\left(\widehat{H}^M\right) + 5\sqrt{2}C \times D_{src}(M) + \Phi(\mathcal{M}) + 2\sqrt{2}C \times \Phi(\mathcal{M},\mathcal{H}) + \frac{6B}{\sqrt{n}} + 3\sqrt{\frac{\log((T+K)/\delta)}{2n}} \quad (10)$$

Note that the high probability bounds in Lemma 2 and Lemma 4 relates to hypothesis class $\mathcal{H}$ only. Therefore, Eq. (10) still holds for $M$ depended on dataset sequence $\{S_t\}_{t=1}^{T+K}$.

### A.2.2  Proof of Proposition 1

$\forall y \in \mathcal{Y}$, we have the following $(*)$:

$$\begin{aligned}
P_{D_{t-1}^W}^{Y=y} &= \int_{\mathcal{X}} P_{D_{t-1}^W}^{X=x,Y=y} dx \\
&= \int_{\mathcal{X}} w_y \times P_{D_{t-1}}^{X=x,Y=y} dx \\
&= \int_{\mathcal{X}} \frac{P_{D_t}^{Y=y}}{P_{D_{t-1}}^{Y=y}} \times P_{D_{t-1}}^{X=x,Y=y} dx \\
&= P_{D_t}^{Y=y} \int_{\mathcal{X}} P_{D_{t-1}}^{X=x|Y=y} dx \\
&= P_{D_t}^{Y=y}
\end{aligned}$$

We have:

$$\begin{aligned}
\mathcal{D}_{KL}\left(P_{D_t}^{X,Y}, P_{D_t^{W,M}}^{X,Y}\right) &= \mathbb{E}_{P_{D_t}^{X,Y}}\left[\log P_{D_t}^{X,Y} - \log P_{D_t^{W,M}}^{X,Y}\right] \\
&= \mathbb{E}_{P_{D_t}^{X,Y}}\left[\log P_{D_t}^Y + \log P_{D_t}^{X|Y}\right] - \mathbb{E}_{P_{D_t}^{X,Y}}\left[\log P_{D_t^{W,M}}^Y + \log P_{D_t^{W,M}}^{X|Y}\right] \\
&= \mathbb{E}_{P_{D_t}^{X,Y}}\left[\log P_{D_t}^Y - \log P_{D_t^{W,M}}^Y\right] + \mathbb{E}_{P_{D_t}^{X,Y}}\left[\log P_{D_t}^{X|Y} - \log P_{D_t^{W,M}}^{X|Y}\right] \\
&\overset{(1)}{=} \mathbb{E}_{P_{D_t}^Y}\left[\mathbb{E}_{P_{D_t}^{X|Y}}\left[\log P_{D_t}^{X|Y} - \log P_{D_t^{W,M}}^{X|Y}\right]\right] \\
&= \mathbb{E}_{P_{D_t}^Y}\left[\mathcal{D}_{KL}\left(P_{D_t}^{X|Y} \parallel P_{D_t^{W,M}}^{X|Y}\right)\right] \quad (11)
\end{aligned}$$

We have $\overset{(1)}{=}$ because $P_{D_t^{W,M}}^Y = m_{t-1}\sharp P_{D_{t-1}^W}^Y = P_{D_{t-1}^W}^Y$ for $m_{t-1}: \mathcal{X} \to \mathcal{X}$ and $P_{D_{t-1}^W}^Y = P_{D_t}^Y$ by $(*)$. For JS-divergence $\mathcal{D}_{JS}$, let $P_{D_t'}^{X,Y} = \frac{1}{2}\left(P_{D_t}^{X,Y} + P_{D_t^{W,M}}^{X,Y}\right)$. Then, we have:

$$\begin{aligned}
&\mathcal{D}_{JS}\left(P_{D_t}^{X,Y} \parallel P_{D_t^{W,M}}^{X,Y}\right) \\
&= \frac{1}{2}\mathcal{D}_{KL}\left(P_{D_t}^{X,Y} \parallel P_{D_t'}^{X,Y}\right) + \frac{1}{2}\mathcal{D}_{KL}\left(P_{D_t^{W,M}}^{X,Y} \parallel P_{D_t'}^{X,Y}\right) \\
&\overset{(2)}{=} \frac{1}{2}\left(\mathbb{E}_{P_{D_t}^Y}\left[\mathcal{D}_{KL}\left(P_{D_t}^{X|Y} \parallel P_{D_t'}^{X|Y}\right)\right] + \mathbb{E}_{P_{D_t^{W,M}}^Y}\left[\mathcal{D}_{KL}\left(P_{D_t^{W,M}}^{X|Y} \parallel P_{D_t'}^{X|Y}\right)\right]\right) \\
&= \mathbb{E}_{P_{D_t}^Y}\left[\frac{1}{2}\left(\mathcal{D}_{KL}\left(P_{D_t}^{X|Y} \parallel P_{D_t'}^{X|Y}\right) + \mathcal{D}_{KL}\left(P_{D_t^{W,M}}^{X|Y} \parallel P_{D_t'}^{X|Y}\right)\right)\right] \\
&= \mathbb{E}_{P_{D_t}^Y}\left[D_{JS}\left(P_{D_t}^{X|Y} \parallel P_{D_t^{W,M}}^{X|Y}\right)\right]
\end{aligned}$$

We have $\overset{(2)}{=}$ by applying Eq. (11) for $\mathcal{D}_{KL}\left(P_{D_t}^{X,Y} \parallel P_{D_t'}^{X,Y}\right)$ and $\mathcal{D}_{KL}\left(P_{D_t^{W,M}}^{X,Y} \parallel P_{D_t'}^{X,Y}\right)$.

### A.2.3 Proof of Proposition 2

First, we show that for any $t \in [1, \cdots, T]$, we have:

$$\mathbb{E}\left[\mathcal{D}_{JS}\left(P_{\widehat{\alpha}_t}^Z \parallel P_{\widehat{\beta}_t}^Z\right)\right] \leq \mathcal{D}_{JS}\left(P_{\alpha_t^*}^Z \parallel P_{\beta_t^*}^Z\right) + \mathcal{O}\left(\left(\frac{1}{\sqrt{n}}\right) \times C(\mathcal{A}, \mathcal{B}, \Gamma)\right) \tag{12}$$

Proposition 2 is then obtained by applying Eq.( 12) for all $t \in [1, \cdots, T]$ followed by averaging over $t$.

**Proof of Eq.( 12).** To simplify the mathematical notation, we omit the index $t$ in the following. Our proof is based on the proof provided for GAN model by Biau et al. [2020]. Let $L(\alpha, \beta, \gamma) = \int_{\mathcal{Z}} \left(\log\left(D_\gamma(z)\right) P_\alpha^z + \log\left(1 - D_\gamma(z)\right) P_\beta^z\right) dz$ and $\widehat{L}(\alpha, \beta, \gamma)$ is the corresponding empirical error, we have:

$$
\begin{aligned}
2\mathcal{D}_{JS}\left(P_{\widehat{\alpha}}^Z \parallel P_{\widehat{\beta}}^Z\right) &= L(\widehat{\alpha}, \widehat{\beta}, \widehat{\gamma}) + \log(4) \\
&\leq \sup_\gamma L(\widehat{\alpha}, \widehat{\beta}, \gamma) + \log(4) \\
&\leq \sup_\gamma \left(\widehat{L}(\widehat{\alpha}, \widehat{\beta}, \gamma) + \left|\widehat{L}(\widehat{\alpha}, \widehat{\beta}, \gamma) - L(\widehat{\alpha}, \widehat{\beta}, \gamma)\right|\right) + \log(4) \\
&\leq \sup_\gamma \widehat{L}(\widehat{\alpha}, \widehat{\beta}, \gamma) + \sup_\gamma \left|\widehat{L}(\widehat{\alpha}, \widehat{\beta}, \gamma) - L(\widehat{\alpha}, \widehat{\beta}, \gamma)\right| + \log(4) \\
&\leq \inf_{\alpha,\beta} \sup_\gamma \widehat{L}(\alpha, \beta, \gamma) + \sup_{\alpha,\beta,\gamma} \left|\widehat{L}(\alpha, \beta, \gamma) - L(\alpha, \beta, \gamma)\right| + \log(4) \\
&\leq \inf_{\alpha,\beta} \sup_\gamma L(\alpha, \beta, \gamma) + \left|\inf_{\alpha,\beta} \sup_\gamma \widehat{L}(\alpha, \beta, \gamma) - \inf_{\alpha,\beta} \sup_\gamma L(\alpha, \beta, \gamma)\right| \\
&\quad + \sup_{\alpha,\beta,\gamma} \left|\widehat{L}(\alpha, \beta, \gamma) - L(\alpha, \beta, \gamma)\right| + \log(4) \\
&\overset{(1)}{\leq} \inf_{\alpha,\beta} \sup_\gamma L(\alpha, \beta, \gamma) + \sup_{\alpha,\beta} \left|\sup_\gamma \widehat{L}(\alpha, \beta, \gamma) - \sup_\gamma L(\alpha, \beta, \gamma)\right| \\
&\quad + \sup_{\alpha,\beta,\gamma} \left|\widehat{L}(\alpha, \beta, \gamma) - L(\alpha, \beta, \gamma)\right| + \log(4) \\
&\overset{(2)}{\leq} \inf_{\alpha,\beta} \sup_\gamma L(\alpha, \beta, \gamma) + 2 \sup_{\alpha,\beta,\gamma} \left|\widehat{L}(\alpha, \beta, \gamma) - L(\alpha, \beta, \gamma)\right| + \log(4) \\
&= 2\mathcal{D}_{JS}\left(P_{\alpha^*}^Z \parallel P_{\beta^*}^Z\right) + 2 \sup_{\alpha,\beta,\gamma} \left|\widehat{L}(\alpha, \beta, \gamma) - L(\alpha, \beta, \gamma)\right|
\end{aligned}
$$

We have $\overset{(1)}{\leq}$ by using inequality $|\inf A - \inf B| \leq \sup |A - B|$, $\overset{(2)}{\leq}$ by using inequality $|\sup A - \sup B| \leq \sup |A - B|$. Take the expectation and rearrange the both sides, we have:

$$\mathbb{E}\left[\mathcal{D}_{JS}\left(P_{\widehat{\alpha}}^{Z} \parallel P_{\widehat{\beta}}^{Z}\right)\right] - \mathcal{D}_{JS}\left(P_{\alpha^*}^{Z} \parallel P_{\beta^*}^{Z}\right)$$

$$\leq \mathbb{E}\left[\sup_{\alpha,\beta,\gamma}\left|\widehat{L}(\alpha,\beta,\gamma) - L(\alpha,\beta,\gamma)\right|\right]$$

$$= \mathbb{E}\left[\sup_{\alpha,\beta,\gamma}\left|\frac{1}{n}\sum_{i=1}^{n}\log\left(D_\gamma((z_t^i))\right) + \frac{1}{n}\sum_{i=1}^{n}\log\left(1 - D_\gamma(z_{t+1}^i)\right)\right.\right.$$
$$\left.\left. - \int_{\mathcal{Z}}\left(\log\left(D_\gamma(z)\right)P_\alpha^z + \log\left(1 - D_\gamma(z)\right)P_\beta^z\right)dz\right|\right]$$

$$\leq \mathbb{E}\left[\sup_{\alpha,\beta,\gamma}\left|\underbrace{\frac{1}{n}\sum_{i=1}^{n}\log\left(D_\gamma((z_t^i))\right) - \int_{\mathcal{Z}}\left(\log\left(D_\gamma(z)\right)P_\alpha^z\right)dz}_{A_s(\alpha,\beta,\gamma)}\right|\right]$$

$$+ \mathbb{E}\left[\sup_{\alpha,\beta,\gamma}\left|\underbrace{\frac{1}{n}\sum_{i=1}^{n}\log\left(1 - D_\gamma(z_{t+1}^i)\right) - \int_{\mathcal{Z}}\left(\log\left(1 - D_\gamma(z)\right)P_\beta^z\right)dz}_{A_t(\alpha,\beta,\gamma)}\right|\right]$$

Note that $(A_s(\alpha,\beta,\gamma))_{\alpha\in\mathcal{A},\beta\in\mathcal{B},\gamma\in\Gamma}$ and $(A_t(\alpha,\beta,\gamma))_{\alpha\in\mathcal{A},\beta\in\mathcal{B},\gamma\in\Gamma}$ are the subgaussian processes in the metric spaces $(\mathcal{A}\times\mathcal{B}\times\Gamma, C_1\|\cdot\|/\sqrt{n})$ and $(\mathcal{A}\times\mathcal{B}\times\Gamma, C_1\|\cdot\|/\sqrt{n})$ where $C_1$ is a constant and $\|\cdot\|$ is the Euclidean norm on $\mathcal{A}\times\mathcal{B}\times\Gamma$. Then using Dudley's entropy integral, we have:

$$\mathbb{E}\left[\mathcal{D}_{JS}\left(P_{\widehat{\alpha}}^{Z} \parallel P_{\widehat{\beta}}^{Z}\right)\right] - \mathcal{D}_{JS}\left(P_{\alpha^*}^{Z} \parallel P_{\beta^*}^{Z}\right)$$

$$\leq \mathbb{E}\left[\sup_{\alpha,\beta,\gamma}A_s(\alpha,\beta,\gamma)\|\right] + \mathbb{E}\left[\sup_{\alpha,\beta,\gamma}A_t(\alpha,\beta,\gamma)\|\right]$$

$$\leq 12\int_0^\infty\left(\sqrt{\log N(\mathcal{A}\times\mathcal{B}\times\Gamma, C\|\cdot\|/\sqrt{n}, \epsilon)} + \sqrt{\log N(\mathcal{A}\times\mathcal{B}\times\Gamma, C\|\cdot\|/\sqrt{n}, \epsilon)}\right)d\epsilon$$

$$= \frac{24C_1}{\sqrt{n}}\int_0^\infty\sqrt{\log N(\mathcal{A}\times\mathcal{B}\times\Gamma, \|\cdot\|, \epsilon)}d\epsilon$$

$$\stackrel{(3)}{=} \frac{24C_1}{\sqrt{n}}\int_0^{\text{diam}(\mathcal{A}\times\mathcal{B}\times\Gamma)}\sqrt{\log N(\mathcal{A}\times\mathcal{B}\times\Gamma, \|\cdot\|, \epsilon)}d\epsilon$$

$$\stackrel{(4)}{\leq} \frac{24C_1}{\sqrt{n}}\int_0^{\text{diam}(\mathcal{A}\times\mathcal{B}\times\Gamma)}\sqrt{\log\left(\left(\frac{2C_2\sqrt{\dim(\mathcal{A}\times\mathcal{B}\times\Gamma)}}{\epsilon}\right)^{\dim(\mathcal{A}\times\mathcal{B}\times\Gamma)}\right)}d\epsilon$$

$$= \mathcal{O}\left(\left(\frac{1}{\sqrt{n}}\right)\times C(\mathcal{A},\mathcal{B},\Gamma)\right)$$

where $\text{diam}(\cdot)$ and $\dim(\cdot)$ are the diameter and the dimension of the metric space, and $C(\mathcal{A},\mathcal{B},\Gamma)$ is the function of $\text{diam}(\mathcal{A}\times\mathcal{B}\times\Gamma)$ and $\dim(\mathcal{A}\times\mathcal{B}\times\Gamma)$. We have $\stackrel{(3)}{=}$ because $N(\mathcal{A}\times\mathcal{B}\times\Gamma, \|\cdot\|, \epsilon) = 1$ for $\epsilon > \text{diam}(\mathcal{A}\times\mathcal{B}\times\Gamma)$, $\stackrel{(4)}{\leq}$ by using inequality $N(\mathcal{T}, \|\cdot\|, \epsilon) \leq \left(\frac{2C_2\sqrt{d}}{\epsilon}\right)^d$ where $\mathcal{T}$ lied in Euclidean space $\mathbb{R}^d$ is the set of vectors whose length is at most $C_2$.

# B MODEL DETAILS

Our proposed model AIRL consists of three components: (i) encoder Enc that maps inputs to representation (i.e., equivalent to $g_t$ in our theoretical results), (ii) transformer layer Trans that helps to enforce the invariant representation (i.e., Enc + Trans equivalent to $f_t$ in our theoretical results), and (iii) classification network LSTM that generates classifiers mapping representations to the output space. At each target domain, LSTM layer is used to generate the new classifier based on the sequences of previous classifiers. The detailed architectures of these networks used in our experiment are presented in Tables 4 and 5 below.

Table 4: Detailed architecture of AIRL for **RMNIST** (**n_channel** = 1, **n_output** = 10), **Yearbook** (**n_channel** = 3, **n_output** = 1), and **CLEAR** (**n_channel** = 3, **n_output** = 10) datasets.

| Networks | Layers |
|---|---|
| Representation Mapping $G$ | Conv2d(input channel = **n_channel**, output channel = 32, kernel = 3, padding = 1) |
| | BatchNorm2d |
| | ReLU |
| | MaxPool2d |
| | Conv2d(input channel = 32, output channel = 32, kernel = 3, padding = 1) |
| | BatchNorm2d |
| | ReLU |
| | MaxPool2d |
| | Conv2d(input channel = 32, output channel = 32, kernel = 3, padding = 1) |
| | BatchNorm2d |
| | ReLU |
| | MaxPool2d |
| | Conv2d(input channel = 32, output channel = 32, kernel = 3, padding = 1) |
| | BatchNorm2d |
| | ReLU |
| | MaxPool2d |
| Transformer Trans | $Q$: Linear(input dim = 32, output dim = 32) |
| | $K$: Linear(input dim = 32, output dim = 32) |
| | $V$: Linear(input dim = 32, output dim = 32) |
| | $U$: Linear(input dim = 32, output dim = 32) |
| | Linear(input dim = 32, output dim = 32) |
| | Batchnorm1d |
| | LeakyReLU |
| Classification Network LSTM | Linear(input dim = (32 * 32 + 32) + (32 * **n_output** + **n_output**), output dim = 128) |
| | LSTM(input dim = 128, output dim = 128) |
| | Linear(input dim = 128, output dim = (32 * 32 + 32) + (32 * **n_output** + **n_output**)) |
| $\widehat{h}_t$ (Output of LSTM) | Linear(input dim = 32, output dim = 32) |
| | ReLU |
| | Linear(input dim = 32, output dim = **n_output**) |

Table 5: Detailed architecture of `AIRL` for **Circle** and **Circle-Hard** datasets.

| Networks | Layers |
|---|---|
| Encoder Enc | Linear(input dim = 2, output dim = 32) |
| | ReLU |
| | Linear(input dim = 32, output dim = 32) |
| | ReLU |
| | Linear(input dim = 32, output dim = 32) |
| | ReLU |
| | Linear(input dim = 32, output dim = 32) |
| Transformer Trans | $Q$: Linear(input dim = 32, output dim = 32) |
| | $K$: Linear(input dim = 32, output dim = 32) |
| | $V$: Linear(input dim = 32, output dim = 32) |
| | $U$: Linear(input dim = 32, output dim = 32) |
| | Linear(input dim = 32, output dim = 32) |
| | Batchnorm1d |
| | LeakyReLU |
| Classification Network LSTM | Linear(input dim = (32 * 32 + 32) + (32 * 1 + 1), output dim = 128) |
| | LSTM(input dim = 128, output dim = 128) |
| | Linear(input dim = 128, output dim = (32 * 32 + 32) + (32 * 1 + 1)) |
| $\widehat{h}_t$ (Output of LSTM) | Linear(input dim = 32, output dim = 32) |
| | ReLU |
| | Linear(input dim = 32, output dim = 1) |

# C  DETAILS OF EXPERIMENTAL SETUP AND ADDITIONAL RESULTS

## C.1  EXPERIMENTAL SETUP

**Datasets.**  Our experiments are conducted on two synthetic and two real-world datasets. The data statistics of these datasets are presented in Table 6. For Eval-S scenario, the first half of domains in the domain sequences are used for training and the following domains are used for testing. For Eval-D scenario, we vary the size of the training set starting from the first half of domains by sequentially adding new domains to this set. In both scenarios, we split the training set into smaller subsets with a ratio $81 : 9 : 10$; these subsets are used as training, validation, and in-distribution testing sets. The data descriptions are given as follow:

- **Circle** [Pesaranghader and Viktor, 2016]: A synthetic dataset containing 30 domains. Features $X := [X_1, X_2]^T$ in domain $t$ are two-dimensional and Gaussian distributed with mean $\bar{X}^t = [r\cos(\pi t/30), r\sin(\pi t/30)]$ where $r$ is radius of semicircle; the distributions of different domains have the same covariance matrix but different means that uniformly evolve from right to left on a semicircle. Binary label $Y$ are generated based on labeling function $Y = \mathbb{1}\left[(X_1 - x_1^o)^2 + (X_2 - x_2^o)^2 \leq r\right]$, where $(x_1^o, x_2^o)$ are center of semicircle. Models trained on the right part are evaluated on the left part of the semicircle.

- **Circle-Hard**: A synthetic dataset adapted from **Circle** dataset, where mean $\bar{X}^t$ does not uniformly evolve. Instead, $\bar{X}^t = [r\cos(\theta_t), r\sin(\theta_t)]$ where $\theta_t = \theta_{t-1} + \pi(t-1)/180$ and $\theta_1 = 0$ rad.

- **RMNIST**: A dataset constructed from MNIST [LeCun et al., 1998] by $R$-degree counterclockwise rotation. We evenly select 30 rotation angles $R$ from $0°$ to $180°$ with step size $6°$; each angle corresponds to a domain. The domains with $R \leq r$ are considered source domains, those with $R > r$ are the target domains used for evaluation. In this dataset, the goal is to train a multi-class classifier on source domains that predicts the digits of images in target.

- **Yearbook** [Ginosar et al., 2015]: A real dataset consisting of frontal-facing American high school yearbook photos from 1930-2013. Due to the evolution of fashion, social norms, and population demographics, the distribution of facial images changes over time. In this dataset, we aim to train a binary classifier using historical data to predict the genders of images in the future.

- **CLEAR** [Lin et al., 2021]: A real dataset built from existing large-scale image collections (YFCC100M) which captures the natural temporal evolution of visual concepts in the real world that spans a decade (2004-2014). In this dataset, we aim to train a multi-class classifier using historical data to predict 10 object types in future images.

Table 6: Data statistics.

|  | Data type | Label type | #instance | #domain |
|---|---|---|---|---|
| Circle | Synthetic | Binary | 30000 | 30 |
| Circle-Hard | Synthetic | Binary | 30000 | 30 |
| RMNIST | Semi-synthetic | Multi | 30000 | 30 |
| Yearbook | Real-world | Binary | 33431 | 84 |
| CLEAR | Real-world | Multi | 29747 | 10 |

**Non-stationary mechanisms in synthetic datasets.**  We note that in synthetic datasets, we precisely known the non-stationary mappings that generate domain sequences.

- **Circle**: A synthetic dataset containing 30 domains. Features $X := [X_1, X_2]^T$ in domain $t$ are two-dimensional and Gaussian distributed with mean $\bar{X}^t = [r\cos(\pi t/30), r\sin(\pi t/30)]$ where $r$ is radius of semicircle; the distributions of different domains have the same covariance matrix but different means that uniformly evolve from right to left on a semicircle. Binary label $Y$ are generated based on labeling function $Y = \mathbb{1}\left[(X_1 - x_1^o)^2 + (X_2 - x_2^o)^2 \leq r\right]$, where $(x_1^o, x_2^o)$ are center of semicircle.
$$\Rightarrow \mathrm{m}_t = \begin{bmatrix} \cos(\pi/30) & -\sin(\pi/30) \\ \sin(\pi/30) & \cos(\pi/30) \end{bmatrix} \forall t \in [1, \cdots, 29]$$

- **Circle-Hard**: A synthetic dataset adapted from **Circle** dataset, where mean $\bar{X}^t$ does not uniformly evolve. Instead, $\bar{X}^t = [r\cos(\theta_t), r\sin(\theta_t)]$ where $\theta_t = \theta_{t-1} + \pi(t-1)/180$ and $\theta_1 = 0$ rad.
$$\Rightarrow \mathrm{m}_t = \begin{bmatrix} \cos(\pi t/180) & -\sin(\pi t/180) \\ \sin(\pi t/180) & \cos(\pi t/180) \end{bmatrix} \forall t \in [1, \cdots, 19]$$

- **RMNIST**: A dataset constructed from MNIST by $R$-degree counterclockwise rotation. We evenly select 30 rotation angles $R$ from 0° to 180° with step size 6°; each angle corresponds to a domain.
$\Rightarrow \mathrm{m}_t = \begin{bmatrix} \cos(6°) & -\sin(6°) \\ \sin(6°) & \cos(6°) \end{bmatrix} \forall t \in [1, \cdots, 29]$

**Baseline methods.** We compare the proposed AIRL with existing methods from related areas, including the followings:

- Empirical risk minimization (ERM): A simple method that considers all source domains as one domain.

- Last domain (LD): A method that only trains model using the most recent source domain.

- Fine tuning (FT): The baseline trained on all source domains in a sequential manner.

- Domain invariant representation learning: Methods that learn the invariant representations across source domains and train a model based on the representations. We experiment with G2DM [Albuquerque et al., 2019], DANN [Ganin et al., 2016], CDANN [Li et al., 2018b], CORAL [Sun and Saenko, 2016], IRM [Arjovsky et al., 2019].

- Data augmentation: We experiment with MIXUP [Zhang et al., 2018] that generates new data using convex combinations of source domains to enhance the generalization capability of models.

- Continual learning: We experiment with EWC [Kirkpatrick et al., 2017], method that learns model from data streams that overcomes catastrophic forgetting issue.

- Continuous domain adaptation: We experiment with CIDA [Wang et al., 2020], an adversarial learning method designed for DA with continuous domain labels.

- Distributionally robust optimization: We experiment with GROUPDRO [Sagawa et al., 2019] that minimizes the worst-case training loss over pre-defined groups through regularization.

- Gradient-based DG: We experiment with FISH [Shi et al., 2022] that targets domain generalization by maximizing the inner product between gradients from different domains.

- Contrastive learning-based DG: We experiment with SELFREG [Kim et al., 2021] that utilizes the self-supervised contrastive losses to learn domain-invariant representation by mapping the latent representation of the same-class samples close together.

- Non-stationary environment DG: We experiment with DRAIN [Bai et al., 2022], TKNets [Zeng et al., 2024b], LSSAE [Qin et al., 2022]. and DDA [Zeng et al., 2023]. DRAIN, DPNET, and DDA focus on domain $D_{T+1}$ only so we use the same model when making predictions for all target domains $\{D_t\}_{t>T}$.

**Evaluation method.** In the experiments, models are trained on a sequence of source domains $\mathcal{D}_{src}$, and their performance is evaluated on target domains $\mathcal{D}_{tgt}$ under two different scenarios: Eval-S and Eval-D.

In the scenario Eval-S, models are trained one time on the first half of domain sequence $\mathcal{D}_{src} = [D_1, D_2, \cdots, D_T]$ and are then deployed to make predictions on the next $K$ domains in the second half of domain sequence $\mathcal{D}_{tgt} = [D_{T+1}, D_{T+2}, \cdots, D_{T+K}]$ $(T+1 \leq K \leq 2T)$. The average and worst-case performances can be evaluated using two matrices $\mathrm{OOD}_{\mathrm{Avg}}$ and $\mathrm{OOD}_{\mathrm{Wrt}}$ defined below.

$$\mathrm{OOD}_{\mathrm{Avg}} = \frac{1}{K}\sum_{k=1}^{K}\mathrm{acc}_{T+k}; \quad \mathrm{OOD}_{\mathrm{Wrt}} = \min_{k \in [K]}\mathrm{acc}_{T+k}$$

where $\mathrm{acc}_{T+k}$ denotes the accuracy of model on target domain $D_{T+k}$.

In the scenario Eval-D, source and target domains are not static but are updated periodically as new data/domain becomes available. This allows us to update models based on new source domains. Specifically, at time step $t \in [T, 2T-K]$, models are updated on source domains $\mathcal{D}_{src} = [D_1, D_2, \cdots, D_t]$ and are used to predict target domains $\mathcal{D}_{tgt} = [D_{t+1}, D_{t+2}, \cdots, D_{t+K}]$. The average and worst-case performances of models in this scenario can be defined as follows.

$$\mathrm{OOD}_{\mathrm{Avg}} = \frac{1}{(T-K+1)K}\sum_{t=T}^{2T-K}\sum_{k=1}^{K}\mathrm{acc}_{t+k}$$
$$\mathrm{OOD}_{\mathrm{Wrt}} = \min_{t \in [T, 2T-K]}\frac{1}{K}\sum_{k=1}^{K}\mathrm{acc}_{t+k}$$

In our experiment, the time step $t$ starts from the index denoting half of the domain sequence.

Table 7: Performances of `DANN` on **RMNIST** dataset.

| Target Domain | 0°-rotated | 15°-rotated | 30°-rotated | 45°-rotated | 60°-rotated |
|---|---|---|---|---|---|
| Model Performance | 51.2 | 59.1 | 70.0 | 69.2 | 53.9 |

Table 8: The average training times (i.e., seconds) of non-stationary DG methods for **Circle**,**Circle-Hard**, **RMNIST**, **Yearbook**, and **CLEAR** datasets.

| | **Circle** | **Circle-Hard** | **RMNIST** | **Yearbook** | **CLEAR** |
|---|---|---|---|---|---|
| AIRL | 32 | 25 | 382 | 749 | 1504 |
| LSSAE | 184 | 175 | 1727 | 1850 | 13287 |
| DRAIN | 460 | 230 | 2227 | 5538 | 1920 |
| TKNets | 18 | 13 | 208 | 448 | 1542 |

**Implementation and training details.** Data, model implementation, and training script are included in the supplementary material. We train each model on each setting with 5 different random seeds and report the average prediction performances. All experiments are conducted on a machine with 24-Core CPU, 4 RTX A4000 GPUs, and 128G RAM.

## C.2 ADDITIONAL EXPERIMENT RESULTS

**Performance gap between in-distribution and out-of-distribution predictions.** This study is motivated based on the assumption that the environment changes over time and that there exist distribution shifts between training and test data. To verify this assumption in our datasets, we compare the performances of `ERM` on in-distribution and out-of-distribution testing sets. Specifically, we show the gaps between the performances of `ERM` measured on the in-distribution (i.e., $ID_{Avg}$) and out-of-distribution (i.e., $OOD_{Avg}$) testing sets under Eval-D scenario (i.e., $K = 5$) in Figure 5.

**Performance of fixed invariant representation learning in conventional and non-stationary DG settings.** A key distinction from non-stationary DG is that the model evolves over the domain sequence to capture non-stationary patterns (i.e., learn invariant representations between two consecutive domains but adaptive across domain sequence). This stands in contrast to the conventional DG [Ganin et al., 2016, Phung et al., 2021] which relies on an assumption that target domains lie on or are near the mixture of source domains, then enforcing fixed invariant representations across all source domains can help to generalize the model to target domains. We argue that this assumption may not hold in non-stationary DG where the target domains may be far from the mixture of source domains resulting in the failure of the existing methods.

To verify this argument, we conduct an experiment on rotated **RMNIST** dataset with `DANN` [Ganin et al., 2016] – a model that learns fixed invariant representations across all domains. Specifically, we create 5 domains by rotating images by 0, 15, 30, 45, and 60 degrees, respectively, and follow leave-one-out evaluation (i.e., one domain is target while the remaining domains are source). Clearly, the setting where the target domain are images rotated by 0 or 60 degrees can be considered as non-stationary domain generalization while other settings can be considered as conventional domain generalization. The performances of DANN with different target domains are shown in Table 7. As we can see, the accuracy drops significantly when the target domain are images rotated by 0 or 60 degrees. This result demonstrates that learning fixed invariant representations across all domains is not suitable for non-stationary DG.

**Computation complexity of non-stationary DG methods.** Compared to existing works for non-stationary DG, our method also shows better computational efficiency. It's because of our effective design to capture non-stationary patterns. Specifically, `LSSAE` and `DRAIN` have more complex architectures and objective functions resulting in much more training time than our method. While `TKNets` has slightly better training time than ours, this model requires storing previous data to make predictions and is not generalized to multiple target domains. To further support our claim, the average training times (i.e., seconds) of these methods for different datasets are shown in Table 8.

**Experimental results for Eval-S scenario.** The prediction performances of `AIRL` and baselines on synthetic (i.e., Circle, Circle-Hard) and real-world (i.e., RMNIST, Yearbook) data under Eval-S scenario are presented in Figure 6 below. In this

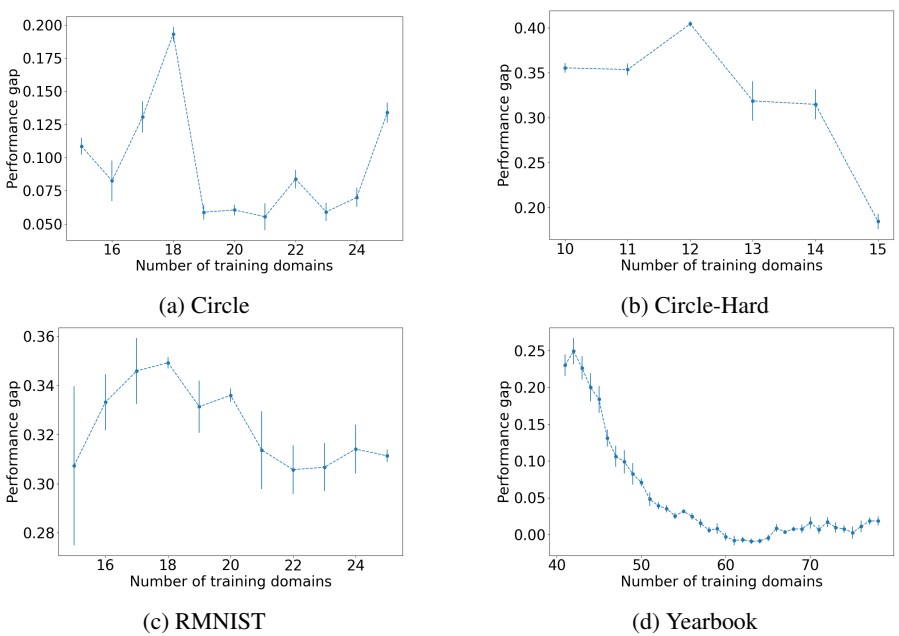

Figure 5: Gaps between the performances of ERM measured on the in-distribution and out-of-distribution testing sets (i.e., $\text{ID}_{\text{Avg}} - \text{OOD}_{\text{Avg}}$) under Eval-D scenario (i.e., $K = 5$). This experiment is conducted on **Circle**, **Circle-Hard**, **RMNIST**, and **Yearbook** datasets.

scenario, the training set is fixed as the first half of domains while the testing set is varied from the five subsequent domains to the second half of domains in the domain sequences. We report averaged results with error bars (std) for training over 5 different random seeds.

We can see that AIRL consistently outperforms baselines in most datasets. We also observe that the prediction performances decreases when the predictions are made for the distant target domains (i.e., the number of testing domain increases) for all models in **Circle**, **Circle-Hard**, and **RMNIST** datasets. This pattern is reasonable because domains in these datasets are generated monotonically. For **Yearbook** dataset, the performance curves are U-shaped that they decrease first but increase later. This dataset is from a real-world environment so we expect the shapes of the curves are more complex compared to those in the other datasets.

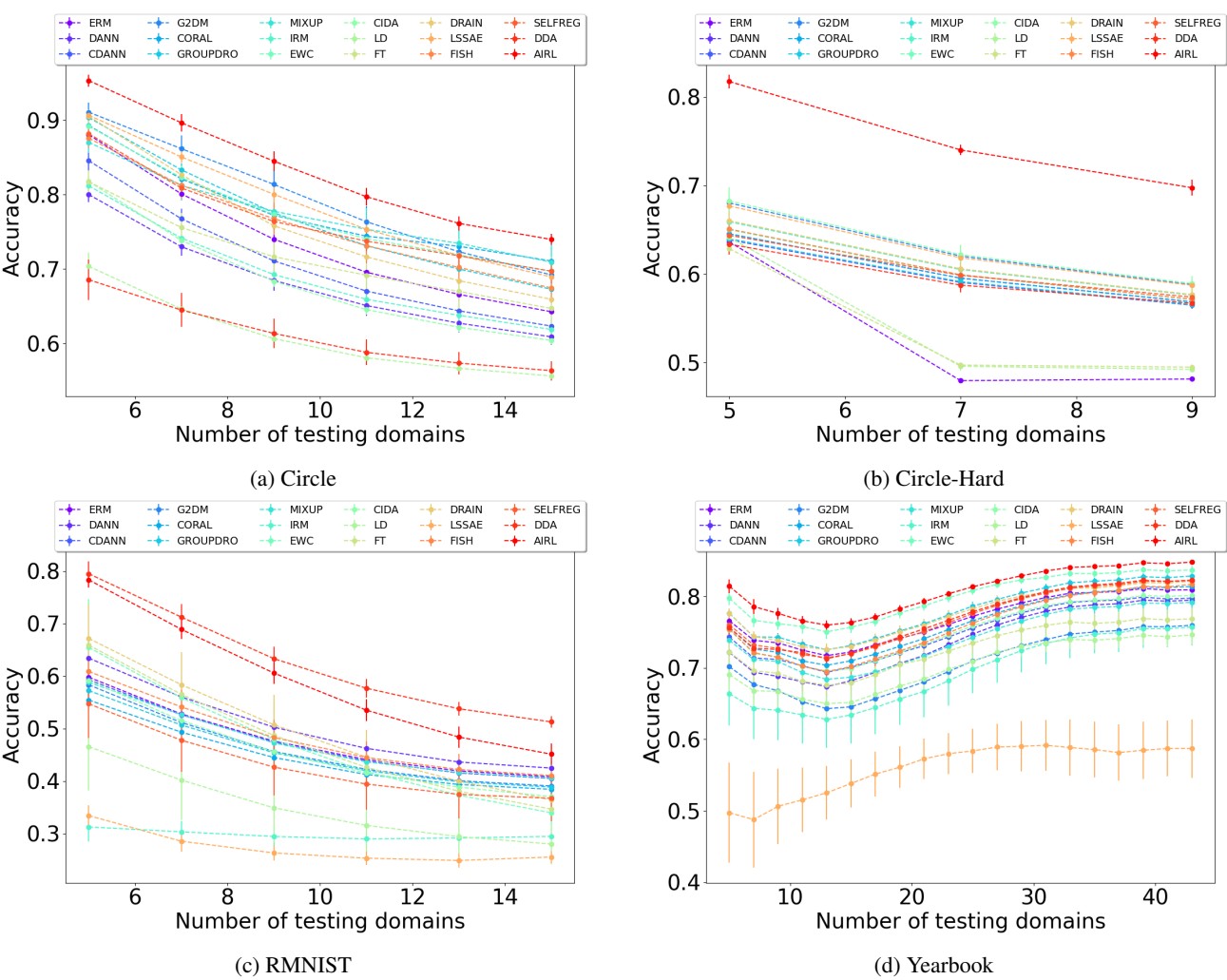

Figure 6: Prediction performances (i.e., $\mathrm{OOD}_{\mathrm{Avg}}$) of AIRL and baselines under Eval-S scenario. The training set is fixed as the first half of domains while the testing set is varied from the five subsequent domains to the second half of domains in the domain sequences. We report average results for training over 5 different random seeds. This experiment is conducted on **Circle**, **Circle-Hard**, **RMNIST**, and **Yearbook** datasets.