# OpenReview forum: "Non-stationary Domain Generalization: Theory and Algorithm"
_auai.org/UAI/2024/Conference — UAI 2024 poster_

### Official Review · Reviewer_Cb27 · 2024-03-11

**Q2-1 Originality-Novelty:** 2
**Q2-2 Correctness-Technical Quality:** 3
**Q2-5 Clarity Of Writing:** 3

**Q10 Ethical Concerns:**

No ethical concerns.

**Q1 Summary And Contributions:**

Summary:
This paper studies non-stationary domain generalization where the training domains evolve over time. Through a comprehensive theoretical study, an upper bound of prediction error on target domains is established. To tackle the non-stationary domain generalization problem, the authors propose a novel method that leverages non-stationary features to learn a robust model that can generalize well to unseen target domains. Experiments on several typical datasets.

**Q2-3 Extent To Which Claims Are Supported By Evidence:**

2: Fair: the main claims are somewhat supported by evidence (but the experimental evaluation may be weak, or does not match entirely with the claims, important baselines may be missing, proofs contain important ideas but lack rigor, algorithmic details are only discussed superficially, references are imprecise, assumptions are not sufficiently motivated or explicated, etc.).

**Q2-4 Reproducibility:**

3: Good: key resources (e.g. proofs, code, data) are available and key details (e.g. proofs, experimental setup) are sufficiently well-described for competent researchers to confidently reproduce the main results.

**Q3 Main Strengths:**

- Extensive theoretical analysis has been provided to justify the proposed method.
- Using LSTM to solve dynamic domain generalization is an interesting idea.
- The experimental performance is quite good, showing superior accuracy to most of the baseline methods.

**Q4 Main Weakness:**

- Unclear motivation for proposing AIRL method. There is no convincing discussion of the vital problem of dynamic domain generalization. How can the proposed design of AIRL can solve such a problem effectively is not clear.
- Insufficient experimental evaluation. Although the theoretical results are quite extensive, however, only quantitative comparison is provided to justify this method. There is no deep understanding of the qualitative aspect of this method.
- Computational efficiency is not discussed. Since the training data is dynamically changing, how fast could the proposed method adapt to the new environment is unknown.

**Q5 Detailed Comments To The Authors:**

Please see weaknesses for detail.

**Q9 Complying With Reviewing Instructions:**

Yes

---

> ### Author Rebuttal · Authors · 2024-04-09
>
> > Q1. Regarding motivation of non-stationary domain generalization and our proposed method (AIRL).
>
> A1. We wish to emphasize that non-stationary domain generalization (DG) poses a significant and challenging problem across diverse fields, including computer vision [1] and healthcare [2]. For instance, consider the healthcare domain, where clinical data evolves over time owing to shifts in disease prevalence. However, researchers typically only have access to historical data to develop predictive models for clinical outcomes, which will be deployed to make predictions on future data. As demonstrated in a previous study [2], these clinical predictive models often exhibit suboptimal performance when applied to future clinical data.
>
> The defining characteristic of non-stationary DG, which sets it apart from conventional DG, is the evolution of data distribution along a specific trajectory (e.g., time, space). This attribute renders non-stationary DG a challenging task, necessitating novel solutions that account for non-stationary mechanisms. We have already conducted experiments to validate this argument (refer to Table 7 in Appendix C.2).
>
> Specifically, we conduct an experiment on rotated RMNIST dataset with DANN [cite] – a conventional DG method that learns invariant representations across all domains. In this experiment, we create 5 domains by rotating images by 0, 15, 30, 45, and 60 degrees, respectively, and follow leave-one-out evaluation (i.e., one domain is target while remaining domains are source). Clearly, the setting where target domain is images rotated by 0 or 60 degrees can be considered as non-stationary DG while other settings can be considered as conventional DG. The performances of DANN with different target domains are shown in the following table (copy from Table 7 in Appendix C.2).
>
> | Target domain    | 0    | 15   | 30   | 45   | 60   |
> |------------------|------|------|------|------|------|
> | DANN performance | 51.2 | 59.1 | 70.0 | 69.2 | 53.9 |
>
> As we can see, the accuracy drops significantly when target domain is images rotated by 0 or 60 degrees. This result demonstrates that conventional DG methods are not suitable for non-stationary DG.
>
> Our proposed method (AIRL) is designed based on insights derived from our theoretical analysis which states that capturing non-stationary is a key in non-stationary DG. Specifically, our model leverages Transformer and LSTM architectures to capture the non-stationary mechanisms that generate domain sequences within the representation space. This design has demonstrated effectiveness across various non-stationary domain generalization datasets (refer to Table 2).
>
> > Q2. Regarding qualitative evaluation.
>
> A2. Thanks for your suggestion. We’ve added a qualitative evaluation in our external PDF file ( https://anonymous.4open.science/r/UAI-2024-Rebuttal-5E07/UAI_2024_Rebuttal.pdf ). We will also include this evaluation in our revised version.
>
> > Q3. Regarding Computational efficiency.
>
> A3. Thanks for your suggestion. We clarify that the design of our method allows it to adapt to multiple target domains without the need of retraining or to access labeled data from target domains. In particular, during inference stage, the predictions for any target domain are made by the trained representation mapping and the domain-specific classifier generated by the trained LSTM (i.e., Algorithm 2 in Appendix B.1). In case labeled target domains are available, fine-tuning can be conducted for better prediction performances.
>
> Compared to existing works for non-stationary domain generalization, our method also shows better computational efficiency. It’s because of our effective design to capture non-stationary patterns. Specifically, LSSAE and DRAIN have more complex architectures and objective functions resulting in much more training time than our method. While DPNET has slightly better training time than ours, this model requires storing previous data to make predictions and is not generalized to multiple target domains. To further support our claim, the average training times (i.e., seconds) of these methods for different datasets are shown in the table below.
>
> |       | Circle | Circle-Hard | RMNIST | Yearbook | CLEAR |
> |-------|--------|-------------|--------|----------|-------|
> | AIRL  | 32     | 25          | 382    | 749      | 1504  |
> | LSSAE | 184    | 175         | 1727   | 1850     | 13287 |
> | DRAIN | 460    | 230         | 2227   | 5538     | 1920  |
> | DPNET | 18     | 13          | 208    | 448      | 1542  |
>
> References
>
> [1] Lin, Zhiqiu, et al. "The clear benchmark: Continual learning on real-world imagery." Thirty-fifth conference on neural information processing systems datasets and benchmarks track (round 2). 2021.
>
> [2] Ji, Christina X., Ahmed M. Alaa, and David Sontag. "Large-scale study of temporal shift in health insurance claims." Conference on Health, Inference, and Learning. PMLR, 2023.

---

### Official Review · Reviewer_Ckef · 2024-03-19

**Q2-1 Originality-Novelty:** 2
**Q2-2 Correctness-Technical Quality:** 3
**Q2-5 Clarity Of Writing:** 3

**Q1 Summary And Contributions:**

This paper studies the domain generation with the assumption that the target domain is non-stationary. To develop the theories and algorithm, the paper assumes the existence of a mechanism modeling the transition of target domains, learned from the source domain. The theories provide a description of this problem's difficulty, based on newly defined quantities measuring non-stationary complexity. To address high-dimensional issues, the paper employs representation learning to capture the transitions of the target domains. Based on theoretical observations, the authors design an algorithm comprising a representation network, used to learn invariant representations, and a classification network for prediction. Extensive experiments are conducted to validate their results.

**Q2-3 Extent To Which Claims Are Supported By Evidence:**

3: Good: the main claims are supported by convincing evidence (in the form of adequate experimental evaluation, proofs, (pseudo-)code, references, assumptions).

**Q2-4 Reproducibility:**

3: Good: key resources (e.g. proofs, code, data) are available and key details (e.g. proofs, experimental setup) are sufficiently well-described for competent researchers to confidently reproduce the main results.

**Q3 Main Strengths:**

- The writing of this paper is good.
- The authors' ideas are formalized clearly and are supported by the proposed theorems.
- The experiments compare the related works, and the presented results validate their findings.

**Q4 Main Weakness:**

- The key assumption that there exists a mechanism modeling the transition of the target domain should be empirically verified.
- It might be helpful to discuss the differences between this work and distribution robust optimization.
- Further exploration of the defined non-stationary complexity would be helpful, for example, by providing its order in relation to the size of the data under some mild assumptions.

**Q5 Detailed Comments To The Authors:**

- Why does the algorithm not utilize the information from received unlabeled data for more precise predictions of the transitions?
- Given that the difficulty of handling non-stationary issues is deferred to learning the transition oracle, validating the rationality of this assumption may be necessary.
- How can the algorithm address non-stationary patterns not encountered in the source domain?
- Can the algorithm update its model on the fly, which might be suitable for tackling new patterns of the transitions?

**Q9 Complying With Reviewing Instructions:**

Yes

---

> ### Author Rebuttal · Authors · 2024-04-09
>
> > Q1. Regarding non-stationary mechanism verification.
>
> A1. Thank you for your suggestion. We would like to clarify that the assumption regarding the existence of a mechanism modeling the transition across domains has been empirically verified in our experiment.
>
> -	For synthetic datasets (i.e., Circle, Circle-Hard, and Rotated MNIST), we precisely know the non-stationary mechanism generating the data. Specifically, data in Circle and Circle-Hard are derived from two-dimensional Gaussian distributions, where the distribution mean evolves from right to left on a semicircle. Meanwhile, data in Rotated MNIST are generated by counterclockwise rotation applied to digit images. The mappings between two consecutive domains remain fixed across the domain sequence for Circle and Rotated MNIST, while they vary (but adhere to certain rules) across the domain sequence for Circle-Hard.
>
> -	For real datasets (i.e., Yearbook and CLEAR), although we do not have an exact understanding of the mechanism generating the data, we know that these datasets are collected over time, with each domain having a distinct corresponding timestamp. It has been demonstrated that there exists an evolution of visual concepts, fashion, social norms, and population demographics over time that changes the distribution of images [1,2].
>
> Additional details regarding these datasets can be found in Appendix C.1 of our paper.
>
> > Q2. Regarding comparison with distributionally robust optimization.
>
> A2. The objective of distributionally robust optimization (DRO) is to learn a model that performs well under the worst-case distribution scenario, aiming for effective generalization to test data. This objective aligns closely with the goal of domain generalization. Consequently, several studies have explored DRO-based approaches for domain generalization [3,4]. However, these methods overlook non-stationary mechanisms, leading to suboptimal prediction performance in non-stationary domain generalization scenarios. This observation is empirically validated through a comparative analysis of our method with GroupDRO [4] across multiple datasets. Table 2 demonstrates that our approach consistently outperforms GroupDRO in non-stationary domain generalization settings. We appreciate your suggestion, and we will incorporate this discussion into our revised version.
>
> > Q3. Regarding non-stationary complexity.
>
> A3. Thanks for your suggestion. Non-stationary complexity quantifies the disparity between source and target domains concerning model classes $\mathcal{M}$ and $\mathcal{H}$. Estimating this term requires access to target data which is not feasible in domain generalization setting. Moreover, this term remains fixed given $\mathcal{M}$ and $\mathcal{H}$.
>
> > Q4. Regarding using unlabeled data.
>
> A4. We would like to clarify that our setting is domain generalization, wherein target domains remain unavailable during training. Therefore, utilizing unlabeled data from target domains are not feasible in this setting.
>
> > Q5. Regarding non-stationary patterns in target domains.
>
> A5. We want to clarify that our method does not assume that non-stationary patterns or mappings remain the same across every pair of consecutive domains. Instead, we operate under the assumption that these mappings are generated by a non-stationary mechanism, and we can estimate this mechanism from the sequence of source domains. To empirically support our claim, we conducted experiments on the Circle-Hard dataset, where the mappings differ across each pair of consecutive domains (but adhere to certain rules). Our experimental results (Table 2) demonstrate that our method significantly outperforms other approaches in this scenario, highlighting the effectiveness of our method for non-stationary domain generalization.
>
> References
>
> [1] Ginosar, Shiry, et al. "A century of portraits: A visual historical record of american high school yearbooks." Proceedings of the IEEE International Conference on Computer Vision Workshops. 2015.
>
> [2] Lin, Zhiqiu, et al. "The clear benchmark: Continual learning on real-world imagery." Thirty-fifth conference on neural information processing systems datasets and benchmarks track (round 2). 2021.
>
> [3] Krueger, David, et al. "Out-of-distribution generalization via risk extrapolation (rex)." International Conference on Machine Learning. PMLR, 2021.
>
> [4] Sagawa, Shiori, et al. "Distributionally Robust Neural Networks." International Conference on Learning Representations. 2020.

---

### Official Review · Reviewer_hijD · 2024-03-22

**Q2-1 Originality-Novelty:** 3
**Q2-2 Correctness-Technical Quality:** 3
**Q2-5 Clarity Of Writing:** 3

**Q1 Summary And Contributions:**

This paper studies the domain generalization problem in non-stationary environments, where the learner can observe $T$ rounds of source data and wants to train models to perform well over $T+K$ rounds without having access to the data from the target domain. The paper first proposes an error bound for the accuracy in the target domain, which is determined by (1) the model's accuracy on the source domain, (2) the accuracy of the distribution evolving model $M$, and (3) the gap between the target and source domains. Based on the theoretical results, this paper provides an algorithm to minimize the first and second terms of the bound. Empirical studies also demonstrate the effectiveness of the proposed methods

**Q2-3 Extent To Which Claims Are Supported By Evidence:**

2: Fair: the main claims are somewhat supported by evidence (but the experimental evaluation may be weak, or does not match entirely with the claims, important baselines may be missing, proofs contain important ideas but lack rigor, algorithmic details are only discussed superficially, references are imprecise, assumptions are not sufficiently motivated or explicated, etc.).

**Q2-4 Reproducibility:**

3: Good: key resources (e.g. proofs, code, data) are available and key details (e.g. proofs, experimental setup) are sufficiently well-described for competent researchers to confidently reproduce the main results.

**Q3 Main Strengths:**

+ The theory for non-stationary domain adaptation is interesting and novel to me
+ The proposed algorithm achieves superior performance in various tasks compared with the previous methods

**Q4 Main Weakness:**

- One of my main concerns is the gap between the theoretical analysis and the proposed methods. The proposed methods are mainly designed to optimize the first and second terms of the bound in Theorem 1. This training process seems more like training the models ($M$ and $H$) to fit the source data rather than generalizing the models to the unseen target domain, as the third and fourth terms are not minimized. Therefore, it is somewhat hard for me to reconcile the strong empirical performance with the theory proposed in this paper.

- From my point of view, the assumption made by this paper may be too pessimistic, as the transition function could be arbitrarily different at each iteration. In the worst case, a model that only fits the source data could perform poorly on the test data.

**Q5 Detailed Comments To The Authors:**

Please refer to the discussion in Q4.

**Q9 Complying With Reviewing Instructions:**

Yes

---

> ### Author Rebuttal · Authors · 2024-04-09
>
> > Q1. Regarding the connection between theoretical analysis and the proposed methods.
>
> A1. We would like to clarify that our proposed algorithm strictly adheres to Theorem 1. Specifically, our theoretical result outlines that the error on target domains is bounded above by four terms: 1) the empirical error on $M$-generated source domains, 2) the average distance between source domains and their corresponding $M$-generated counterparts, 3) the non-stationary complexity term, and 4) the sample complexity term.
>
> -	The fourth term reflects the uncertainty inherent in learning from finite data and remains constant given the training data. Essentially, this term diminishes only as the size of training data increases (i.e., as $n$ grows larger).
> -	Similarly, the third term quantifies the disparity between source and target domains concerning model classes $\mathcal{M}$ and $\mathcal{H}$ and remains fixed given $\mathcal{M}$ and $\mathcal{H}$. In practice, an appropriate design of $\mathcal{M}$ and $\mathcal{H}$ will make this term small. In case this term is large, due to no access to target domains, estimating target domains from training data becomes exceedingly challenging. In practice, our design utilizing Transformer and LSTM to represent $\mathcal{M}$ and $\mathcal{H}$ demonstrates good performance across diverse datasets.
>
> In essence, the third and fourth terms remain constant given the training data and the choice of model classes. Thus, Theorem 1 advocates optimizing the first and second terms to mitigate empirical error on target domains. It is noteworthy that optimizing these two terms also implies that the model not only minimizes prediction error on source domains but also learns non-stationary mechanism from the sequence of source domains (i.e., by minimizing $D\_{src}(M)$), such that this mechanism generalizes well to target domains.
>
> > Q2. Regarding assumption about non-stationary transition functions.
>
> A2. We would like to emphasize that our setting is domain generalization, wherein target domains remain unavailable during training. Without the access to target domains, we rely on an assumption to ensure the feasibility of estimating these domains from the training data. In particular, we expect that the non-stationary mechanism learned from the sequence of source domains will yield mappings that effectively estimate the target domains (i.e., non-stationary complexity is small). In the worst case where this assumption proves invalid, not only our method but also other existing methods will struggle to perform well on the test data. This is because estimating target domains from source domains becomes exceedingly challenging in such circumstances.

---

### Official Review · Reviewer_qFsB · 2024-03-23

**Q2-1 Originality-Novelty:** 3
**Q2-2 Correctness-Technical Quality:** 3
**Q2-5 Clarity Of Writing:** 3

**Q1 Summary And Contributions:**

This work studies the non-stationary domain generalization. The authors propose theoretical models for the impacts of environmental non-stationarity on the model performance and new algorithms based on adaptive invariant representation learning to achieve better performance on tarte domains. They also conduct extensive experiments to verify the effectiveness of the proposed algorithm.

**Q2-3 Extent To Which Claims Are Supported By Evidence:**

3: Good: the main claims are supported by convincing evidence (in the form of adequate experimental evaluation, proofs, (pseudo-)code, references, assumptions).

**Q2-4 Reproducibility:**

3: Good: key resources (e.g. proofs, code, data) are available and key details (e.g. proofs, experimental setup) are sufficiently well-described for competent researchers to confidently reproduce the main results.

**Q3 Main Strengths:**

1. This work differentiates from previous studies with a focus on the non-stationary domain generalization.

2. The authors present a solid theoretical study.

3. The method is novel and effective.

4. The experiments are comprehensive.

**Q4 Main Weakness:**

It would be better with more clarifications to the experiments.

**Q5 Detailed Comments To The Authors:**

It would be better with more clarifications to the experiments:
- Why there is a huge performance gap between the performances of previous methods (e.g., LSSAE) in table 2 than those reported in the paper?
- A closely related work misses discussion in the paper [1]
- It'd be better if some visualizations could be provided for some datasets such as Circle to understand the advantages of AIRL.



[1] Enhancing Evolving Domain Generalization through Dynamic Latent Representations, AAAI'24.

**Q9 Complying With Reviewing Instructions:**

Yes

---

> ### Author Rebuttal · Authors · 2024-04-09
>
> > Q1. Regarding performance of LSSAE.
>
> A1. The implementation of LSSAE in our work is adapted from the authors' code available at (https://github.com/WonderSeven/LSSAE). In comparison to the authors' version, we substituted the original encoder network with our own encoder network to ensure a fair comparison with other baselines, ensuring uniformity across all models employing the same backbone. We have noticed that training LSSAE is unstable for image datasets, and the model's performance appears to be sensitive to variations in image encoder architecture, as also indicated in the authors' implementation (https://github.com/WonderSeven/LSSAE/blob/main/network/model_func.py#L38). Additionally, differences in experimental settings may contribute to performance disparities. We have employed a distinct approach to constructing training, validation, and testing sets compared to the methodology described in the LSSAE paper.
>
> > Q2. Regarding related work.
>
> A2. Thanks for pointing out this interesting work. This work proposed a method for non-stationary domain generalization named Information-Based Sequential Autoencoders (MISTS). Similar to LSSAE, MISTS also adopts a variational inference strategy to identify the underlying invariant and dynamic features. However, this paper proposed a new information-theoretic objective that have been shown to be more effective than the one used in LSSAE. Compared to both LSSAE and MISTS, our work contributes not only an effective method but also a theoretical analysis for non-stationary domain generalization. We will refer to this paper and add this discussion in our revised version. Since this paper is published in AAAI 2024, we consider it as the concurrent work.
>
> > Q3. Regarding visualization.
>
> A3. Thanks for your suggestion. We’ve added visualization in our external PDF file ( https://anonymous.4open.science/r/UAI-2024-Rebuttal-5E07/UAI_2024_Rebuttal.pdf )
> We will also add this visualization in our revised version.

---

### Official Review · Reviewer_bfzb · 2024-03-25

**Q2-1 Originality-Novelty:** 3
**Q2-2 Correctness-Technical Quality:** 3
**Q2-5 Clarity Of Writing:** 2

**Q1 Summary And Contributions:**

The paper considers the problem of *nonstationary* domain generalization (DG), i.e., the task of learning from multiple labelled source domains with the goal of generalizing to unseen target domains. In contrast to the standard DG setting where domains are an unstructured set of distributions, in nonstationary DG domains differ systematically along a particular axis $t$ such as time or space. This is formalized through a set of related mechanisms $...., m_{t-1}, m_t, ... $ which generate domain  $t$ (i.e., joint distribution $P^{X,Y}_{t}$) from the previous one.

The paper analyses the nonstationary DG setting from a theoretical perspective (Sec. 4) and provides a generalization bound (Thm.1) which depends on the empirical source domain error, the complexity of the model class and nostationarity pattern, and the sample size. The paper then proposes a method and algorithm for nonstationary DG (Sec. 5) that is inspired by the theoretical analysis. The main ingredients are a representation network consisting of an encoder and a transformer, which aims to learn representation that are aligned between pairs of consequtive domains; and a classification network operating on these representations, whose parameters are updated recurrently using an LSTM. Extensive experiments (Sec. 6) evaluate the proposed method (AIRL) against numerous baselines on several synthetic and real datasets, where AIRL performs favourably. An ablation study verifies that dropping any component of the proposed architecture hurts performance.

**Q2-3 Extent To Which Claims Are Supported By Evidence:**

4: Excellent: all claims are supported by very convincing evidence (in the form of comprehensive experimental evaluation, rigorous mathematical proofs, detailed (pseudo-)code, precise references, well-motivated and realistic assumptions) and the authors deliver what they promise.

**Q2-4 Reproducibility:**

3: Good: key resources (e.g. proofs, code, data) are available and key details (e.g. proofs, experimental setup) are sufficiently well-described for competent researchers to confidently reproduce the main results.

**Q3 Main Strengths:**

- The considered problem setting is an interesting and non-standard variation of the standard DG setting, which appears to be quite relevant for practical applications.
- The paper presents non-trivial theoretical and algorithmic/methodological contributions.
- The empirical evaluation is very thorough, considering a large number of relevant baselines on several datasets, including high-dimensional, non-synthetic data.
- The proposed method is shown to outperform alternatives across most settings, and is theoretically grounded, making it an interesting addition to the literature and potentially a useful tool for nonstationary DG problems in practice.

**Q4 Main Weakness:**

- The main weakness of this paper is the (lack of) clarity of writing and presentation: especially in the first half of the paper (Sec. 1-4) it is not made sufficiently clear exactly how nonstationary DG differs from other types of distribution shift, and why we should care about this setting. Adding more intuition and motivating examples (see detailed comments below for suggestion) would help make the paper more accessible and easier to follow. Another key source of confusion is the introduction of a (sequence of) mechanisms that relate each domain to the next: this seems to be an abstract object (mapping between joint distributions) and it not clear why this is necessary/helpful, or how it relates to the practical implementation.
- There are several grammatical errors (mostly missing articles "the" / "a") which unnecessarily hamper the flow and make the paper more difficult to read/follow. These could (and should) be easily fixed.

**Q5 Detailed Comments To The Authors:**

Despite some difficulty in following some parts, I quite liked the paper and think that is is overall well-structured and provides valuable contributions. As written above, I think the main weakness in its current form is the presentation, which could be substantially improved.

### Suggestions to improve clarity/accessibility
My main suggestion to this end would be to focus the introduction more on the differences between standard and nonstationary DG, e.g., by emphasizing that **``in vanilla DG, the domains are an unordered set, whereas in nonstationary DG, they are an ordered tuple with (temporal/discrete) structure''**. At least from my perspective, this seems to be the main difference, but any further insights/intuition you have on how this differs from other/more general types of domain shifts (e.g., causally structured ones) would be helpful to add here.

In addition to the above, I feel that it would be very helpful to **add a Figure to the begging of the paper which illustrates this setting**. I think the RotatedMNIST dataset could be a good example for this, where a representative image from each domain is shown, highlighting the domain drift (rotation angle) over time as we go from t=1, ..., T from left to right.

Further, it would be helpful to not only have a visual, but also an **algebraic motivating example**, which gives an example for some of the key objects (e.g., the different types of mechanisms $M, m_t, \mathcal{M}, \mathbb{M}$) in a specific context such as the rotation one.

### Further comments and questions:
- The definition of domains in the Notation paragraph in Sec. 3 seems to implicitly assume variables are discrete: for continuous $X$, the input should be a sigma algebra, or if talking about the density, the output can be larger than $1$. (I don't think this level of detail is necessary here, but if included, it should be rigorous and correct in general.)
- When does Assumption 1 hold or not hold for common loss functions? I think MSE would violate it, but I guess you only consider classification tasks? Is cross entropy bounded, and, if so, what is the constant C in this case?
- Could you add some intuition on Assumption 2? E.g., for which types of distributions/models does it (tend to) hold?
- What type of object is the (meta-)mechanism $\mathbb{M}$? Is it a distribution over functions/a conditional distribution of distributions? Giving some examples (or reducing the notation/complexity if possible) would be helpful here.
- How tight is the bound in Theorem 1? How predictive is it of the empirical results? It would be nice to connect this to some of the simulations if possible, or otherwise comment on why this is not feasible.
- Generally, adding a discussion section that talks about limitations/challenges/shortcomings would be appreciated.
- I believe the push-forward notation is used incorrectly in the context of the distributions over $Z$ induced by $f,g$: if $Z=f(X)$, then $P^Z=f_* P^X$, but the paper talks about $f_*P^Z$ (and similarly for $g$), which does not seem to make sense/is not well defined if $f$ takes $X$ as an input.
- On page 6, 2nd column, it is stated that Trans operates on $[z^j_1, ..., z^j_t]$. Why is this meaningful? Do you assume that samples are aligned across domains/come in the form of time-series? Else, why would there be any relation between the $j$th sample in domain $t-1$ and the jth sample in domain $t$?
- There is some inconsistency in the notation (t vs t+1) in the sentence after Eq. (2).
- Page 6, middle paragraph: isn't pairwise alignment equivalent to global alignment, at least theoretically (i.e., at the global optimum)? That is, if $D_{t-1}$ & $D_t$ are aligned, and $D_t$ & $D_{t+1}$ are aligned, doesn't this imply that $D_{t-1}$ & $D_{t+1}$ are also aligned? If so, in which sense is the pairwise objective different from seeking globally invariant representations, is this about approximate alignment/invariance?
- In Eq. (3), why is the first expectation over the reweighted version of $D_t$? Please add an explanation.
- The expressions for the covariance matrices in (5) and (6) don't add much (AFAIK, this is not the standard/typical way of writing the covariance matrix). I think this space would be better spent on adding intuition (e.g., on what this objective does).
- I would like to see at least some short discussion of the Experiments on the static (EVAL-S) setting in the main paper---how come there is no large table mirroring Table 2 in the appendix for EVAL-S?

**Q9 Complying With Reviewing Instructions:**

Yes

---

> ### Author Rebuttal · Authors · 2024-04-09
>
> > Q1. Regarding clarity about difference between non-stationary and conventional DG
>
> Thank you very much for your valuable suggestions. We’ve incorporated new figure to compare these 2 DG settings, along with an algebraic example for rotation case in our external PDF file ( https://anonymous.4open.science/r/UAI-2024-Rebuttal-5E07/UAI_2024_Rebuttal.pdf ). We will enhance introduction of non-stationary DG and integrate new materials into revised version.
>
> > Q2. Regarding definition of random variable (RV)
>
> You’re right. To address continuous RV, it's essential to place them within context of sigma algebra. The reason we implicitly treat RV as discrete is to simplify our notations. However, it's important to note that all our theoretical results extend to scenarios involving continuous RV.
>
> > Q3. Regarding Assumption 1
>
> Cross-entropy loss is unbounded. However, we can modify cross-entropy loss to make it satisfied Assumption 1. In particular, it can be bounded by $C$ by modifying softmax output from $\left(p_1,\cdots,p_{|\mathcal{Y}|}\right)$ to $\left(\hat{p}\_1,\cdots,\hat{p}\_{|\mathcal{Y}|}\right)$ where $\hat{p}_i=p_i \left(1-exp\left(-C \right)\left|\mathcal{Y}\right|\right)+exp\left(-C\right)$.
>
> > Q4. Regarding Assumption 2
>
> According to [1] (Theorem 11 page 82), Assumption 2 holds when input space is compact and bounded in unit $L\_2$ ball and function $f$ in $\mathcal{L}\_{\mathcal{H}}$ is linear and $R$-Lipschitz in $l_2$ norm.
>
> > Q5. Regarding non-stationary mechanism
>
> $\mathbb{M}$ is the concept that generates sequence of mappings $\mathbb{m}\_1,\cdots,\mathbb{m}\_t$ across domain sequence. In particular, $\mathbb{m}\_t$ is autoregressively generated from $\mathbb{m}\_1,\cdots,\mathbb{m}\_{t-1}$ using $\mathbb{M}$. In our algorithms, we instantiate $\mathbb{M}$ using Transformer and LSTM.
>
> > Q6. Regarding tightness of Theorem 1
>
> Estimating tightness of upper bound in Theorem 1 is challenging. This is due to the dependence of this bound on a non-stationary complexity term defined in context of infinite data.
>
> > Q7. Regarding limitations
>
> We acknowledge certain limitations in our work. Regarding theoretical analysis, we presently lack an effective method to estimate non-stationary complexity from finite data. Concerning algorithm design, our method is unable to address scenarios where data from all source domains are not simultaneously available during training (i.e., online learning). We will incorporate this discussion into revised version.
>
> > Q8. Regarding push-forward notation
>
> You are correct regarding push-forward notation for distribution over $Z$. Initially, we utilized $f \sharp P^X\_D$ as you mentioned. However, recognizing a potential complexity of our math notation that could hinder its association with distribution over $Z$, we have chosen to employ $f \sharp P^Z_D$ instead. This notation clearly emphasizes that it’s related to $Z$.
>
> > Q9. Regarding Trans operation
>
> We do not assume data are aligned across domain sequence. In particular, due to randomness in data loading, there’s no alignment between the j-th sample in domain $D_{t-1}$  and the j-th sample in domain $D_t$. In our original design, we compute $\widehat{z}_t^j$ from all data of previous domains in the batch. However, this approach is computationally expensive. Instead of using all data in the batch, we found that computing $\widehat{z}_t^j$ utilizing only one data point from each previous domain give us fast computation without sacrificing performance. We will clarify it in revised version.
>
> > Q10. Regarding domain index
>
> We have reviewed the sentence following Eq. (2) and confirmed it is correct. However, we acknowledge that the subsequent sentence is incorrect and may cause confusion. The corrected one is “In particular, $z\_j^{t+1}=g\_t\left(x\_j^{t+1}\right)$ and $\widehat{z}\_j^{t}=f\_t\left(x\_j^{t}\right)$”.
>
> > Q11. Regarding pairwise and global invariance
>
> Pairwise invariance doesn’t imply global invariance. It is because we use distinct mappings for different pairs of domains. In particular, $D\_{t-1}$ and $D\_{t}$ are aligned by two mappings $f\_{t-1}$ and $g\_{t-1}$ while $D\_{t}$ and $D\_{t+1}$ are aligned by two mappings $f\_{t}$ and $g\_{t}$.
>
> > Q12. Regarding Eq. (3)
>
> During training, predictions for $D_t$ are generated after reweighting this domain, this’s why the first expectation in Eq. (3) is over $D^W_t$.
>
> > Q13. Regarding covariance matrices in Eq. (5) and (6)
>
> Aligning covariance matrices has been demonstrated to be a simple yet effective method for aligning two distributions [2]. Thus, we employ this approach to align 2 consecutive domains. We will incorporate this discussion into revised version.
>
> > Q14. Regarding Eval-S
>
> Results for Eval-S share similar trend with Eval-D. We will add the discussion as well as the corresponding table into revised version.
>
> References
>
> [1] https://web.stanford.edu/class/cs229t/notes.pdf
>
> [2] Deep coral: Correlation alignment for deep domain adaptation

---

### Meta-Review · Area_Chair_3PWG · 2024-04-22

The paper considers the problem of nonstationary domain generalization, where the domains are related by a set of related mechanisms.
Under this new setting, this paper proposes a theoretical analysis and a detailed generalization bound, and then develops a practical algorithm for nonstationary domain generalization that is inspired by the theoretical analysis. Extensive experiments were conducted to verify the effectiveness of the proposed algorithm.

All reviewers acknowledged the significance of the theoretical contributions and agreed that the experimental results comprehensively validate the proposed algorithm's effectiveness. At the same time, reviewers also noticed that the clarity of the presentation can be improved.